# Fundamental Limits of Online and Distributed Algorithms for Statistical Learning and Estimation

**Ohad Shamir**
Weizmann Institute of Science
`ohad.shamir@weizmann.ac.il`

## Abstract

Many machine learning approaches are characterized by information constraints on how they interact with the training data. These include memory and sequential access constraints (e.g. fast first-order methods to solve stochastic optimization problems); communication constraints (e.g. distributed learning); partial access to the underlying data (e.g. missing features and multi-armed bandits) and more. However, currently we have little understanding how such information constraints fundamentally affect our performance, independent of the learning problem semantics. For example, are there learning problems where any algorithm which has small memory footprint (or can use any bounded number of bits from each example, or has certain communication constraints) will perform worse than what is possible without such constraints? In this paper, we describe how a single set of results implies positive answers to the above, for several different settings.

## 1 Introduction

Information constraints play a key role in machine learning. Of course, the main constraint is the availability of only a finite data set to learn from. However, many current problems in machine learning can be characterized as learning with *additional* information constraints, arising from the manner in which the learner may interact with the data. Some examples include:

- *Communication constraints in distributed learning:* There has been much recent work on learning when the training data is distributed among several machines. Since the machines may work in parallel, this potentially allows significant computational speed-ups and the ability to cope with large datasets. On the flip side, communication rates between machines is typically much slower than their processing speeds, and a major challenge is to perform these learning tasks with minimal communication.

- *Memory constraints:* The standard implementation of many common learning tasks requires memory which is super-linear in the data dimension. For example, principal component analysis (PCA) requires us to estimate eigenvectors of the data covariance matrix, whose size is quadratic in the data dimension and can be prohibitive for high-dimensional data. Another example is kernel learning, which requires manipulation of the Gram matrix, whose size is quadratic in the number of data points. There has been considerable effort in developing and analyzing algorithms for such problems with reduced memory footprint (e.g. [20, 7, 27, 24]).

- *Online learning constraints:* The need for fast and scalable learning algorithms has popularised the use of online algorithms, which work by sequentially going over the training data, and incrementally updating a (usually small) state vector. Well-known special cases include gradient descent and mirror descent algorithms. The requirement of sequentially passing over the data can be seen as a type of information constraint, whereas the small state these algorithms often maintain can be seen as another type of memory constraint.

- *Partial-information constraints:* A common situation in machine learning is when the available data is corrupted, sanitized (e.g. due to privacy constraints), has missing features, or is otherwise partially accessible. There has also been considerable interest in online learning with partial information, where the learner only gets partial feedback on his performance. This has been used to model various problems in web advertising, routing and multiclass learning. Perhaps the most well-known case is the multi-armed bandits problem with many other variants being developed, such as contextual bandits, combinatorial bandits, and more general models such as partial monitoring [10, 11].

Although these examples come from very different domains, they all share the common feature of information constraints on how the learning algorithm can interact with the training data. In some specific cases (most notably, multi-armed bandits, and also in the context of certain distributed protocols, e.g. [6, 29]) we can even formalize the price we pay for these constraints, in terms of degraded sample complexity or regret guarantees. However, we currently lack a general information-theoretic framework, which directly quantifies how such constraints can impact performance. For example, are there cases where any online algorithm, which goes over the data one-by-one, must have a worse sample complexity than (say) empirical risk minimization? Are there situations where a small memory footprint provably degrades the learning performance? Can one quantify how a constraint of getting only a few bits from each example affects our ability to learn?

In this paper, we make a first step in developing such a framework. We consider a general class of learning processes, characterized only by information-theoretic constraints on how they may interact with the data (and independent of any specific problem semantics). As special cases, these include online algorithms with memory constraints, certain types of distributed algorithms, as well as online learning with partial information. We identify cases where any such algorithm must perform worse than what can be attained without such information constraints. The tools developed allows us to establish several results for specific learning problems:

- We prove a new and generic regret lower bound for partial-information online learning with expert advice, of the form $\Omega(\sqrt{(d/b)T})$, where $T$ is the number of rounds, $d$ is the dimension of the loss/reward vector, and $b$ is the number of bits $b$ extracted from each loss vector. It is optimal up to log-factors (without further assumptions), and holds no matter what these $b$ bits are – a single coordinate (as in multi-armed bandits), some information on several coordinates (as in semi-bandit feedback), a linear projection (as in bandit linear optimization), some feedback signal from a restricted set (as in partial monitoring) etc. Interestingly, it holds even if the online learner is allowed to adaptively choose which bits of the loss vector it can retain at each round. The lower bound quantifies directly how information constraints in online learning degrade the attainable regret, independent of the problem semantics.

- We prove that for some learning and estimation problems - in particular, sparse PCA and sparse covariance estimation in $\mathbb{R}^d$ - no online algorithm can attain statistically optimal performance (in terms of sample complexity) with less than $\tilde{\Omega}(d^2)$ memory. To the best of our knowledge, this is the first formal example of a *memory/sample complexity* trade-off in a statistical learning setting.

- We show that for similar types of problems, there are cases where no distributed algorithm (which is based on a non-interactive or serial protocol on i.i.d. data) can attain optimal performance with less than $\tilde{\Omega}(d^2)$ communication per machine. To the best of our knowledge, this is the first formal example of a *communication/sample complexity* trade-off, in the regime where the communication budget is larger than the data dimension, and the examples at each machine come from the same underlying distribution.

- We demonstrate the existence of (synthetic) stochastic optimization problems where any algorithm which uses memory linear in the dimension (e.g. stochastic gradient descent or mirror descent) cannot be statistically optimal.

**Related Work**

In stochastic optimization, there has been much work on lower bounds for sequential algorithms (e.g. [22, 1, 23]). However, these results all hold in an *oracle model*, where data is assumed to be made available in a specific form (such as a stochastic gradient estimate). As already pointed out in

[22], this does not directly translate to the more common setting, where we are given a dataset and wish to run a simple sequential optimization procedure.

In the context of distributed learning and statistical estimation, information-theoretic lower bounds were recently shown in the pioneering work [29], which identifies cases where communication constraints affect statistical performance. These results differ from ours (in the context of distributed learning) in two important ways. First, they pertain to parametric estimation in $\mathbb{R}^d$, where the communication budget per machine is much smaller than what is needed to even specify the answer with constant accuracy ($\mathcal{O}(d)$ bits). In contrast, our results pertain to simpler detection problems, where the answer requires only $\mathcal{O}(\log(d))$ bits, yet lead to non-trivial lower bounds even when the budget size is much larger (in some cases, much larger than $d$). The second difference is that their work focuses on distributed algorithms, while we address a more general class of algorithms, which includes other information-constrained settings. Strong lower bounds in the context of distributed learning have also been shown in [6], but they do not apply to a regime where examples across machines come from the same distribution, and where the communication budget is much larger than what is needed to specify the output.

There are well-known lower bounds for multi-armed bandit problems and other online learning with partial-information settings. However, they crucially depend on the semantics of the information feedback considered. For example, the standard multi-armed bandit lower bound [5] pertain to a setting where we can view a single coordinate of the loss vector, but doesn't apply as-is when we can view more than one coordinate (e.g. [4, 25]), get side-information (e.g. [19]), receive a linear or non-linear projection (as in bandit linear and convex optimization), or receive a different type of partial feedback (e.g. partial monitoring [11]). In contrast, our results are generic and can directly apply to any such setting.

Memory and communication constraints have been extensively studied within theoretical computer science (e.g. [3, 21]). Unfortunately, almost all these results pertain to data which was either adversarially generated, ordered (in streaming algorithms) or split (in distributed algorithms), and do not apply to statistical learning tasks, where the data is drawn i.i.d. from an underlying distribution. [28, 15] do consider i.i.d. data, but focus on problems such as detecting graph connectivity and counting distinct elements, and not learning problems such as those considered here. Also, there are works on provably memory-efficient algorithms for statistical problems (e.g. [20, 7, 17, 13]), but these do not consider lower bounds or provable trade-offs.

Finally, there has been a line of works on hypothesis testing and statistical estimation with finite memory (see [18] and references therein). However, the limitations shown in these works apply when the required precision exceeds the amount of memory available. Due to finite sample effects, this regime is usually relevant only when the data size is exponential in the memory size. In contrast, we do not rely on finite precision considerations.

## 2 Information-Constrained Protocols

We begin with a few words about notation. We use bold-face letters (e.g. $\mathbf{x}$) to denote vectors, and let $\mathbf{e}_j \in \mathbb{R}^d$ denote $j$-th standard basis vector. When convenient, we use the standard asymptotic notation $\mathcal{O}(\cdot), \Omega(\cdot), \Theta(\cdot)$ to hide constants, and an additional ˜ sign (e.g. $\tilde{\mathcal{O}}(\cdot)$) to also hide log-factors. $\log(\cdot)$ refers to the natural logarithm, and $\log_2(\cdot)$ to the base-2 logarithm.

Our main object of study is the following generic class of information-constrained algorithms:

**Definition 1** (($b, n, m$) Protocol). *Given access to a sequence of $mn$ i.i.d. instances (vectors in $\mathbb{R}^d$), an algorithm is a ($b, n, m$) protocol if it has the following form, for some functions $f_t$ returning an output of at most $b$ bits, and some function $f$:*

- *For $t = 1, \ldots, m$*

    - *Let $X^t$ be a batch of $n$ i.i.d. instances*
    - *Compute message $W^t = f_t(X^t, W^1, W^2, \ldots W^{t-1})$*

- *Return $W = f(W^1, \ldots, W^m)$*

Note that the functions $\{f_t\}_{t=1}^m, f$ are completely arbitrary, may depend on $m$ and can also be randomized. The crucial assumption is that the outputs $W^t$ are constrained to be only $b$ bits.

As the definition above may appear quite abstract, let us consider a few specific examples:

- *b-memory online protocols*: Consider any algorithm which goes over examples one-by-one, and incrementally updates a state vector $W^t$ of bounded size $b$. We note that a majority of online learning and stochastic optimization algorithms have bounded memory. For example, for linear predictors, most gradient-based algorithms maintain a state whose size is proportional to the size of the parameter vector that is being optimized. Such algorithms correspond to $(b, n, m)$ protocols where $W^t$ is the state vector after round $t$, with an update function $f_t$ depending only on $W^{t-1}$, and $f$ depends only on $W^m$. $n = 1$ corresponds to algorithms which use one example at a time, whereas $n > 1$ corresponds to algorithms using mini-batches.

- *Non-interactive and serial distributed algorithms:* There are $m$ machines and each machine receives an independent sample $X^t$ of size $n$. It then sends a message $W^t = f_t(X^t)$ (which here depends only on $X^t$). A centralized server then combines the messages to compute an output $f(W^1 \ldots W^m)$. This includes for instance divide-and-conquer style algorithms proposed for distributed stochastic optimization (e.g. [30]). A serial variant of the above is when there are $m$ machines, and one-by-one, each machine $t$ broadcasts some information $W^t$ to the other machines, which depends on $X^t$ as well as previous messages sent by machines $1, 2, \ldots, (t-1)$.

- *Online learning with partial information:* Suppose we sequentially receive $d$-dimensional loss vectors, and from each of these we can extract and use only $b$ bits of information, where $b \ll d$. This includes most types of bandit problems [10].

In our work, we contrast the performance attainable by *any* algorithm corresponding to such a protocol, to *constraint-free* protocols which are allowed to interact with the data in any manner.

## 3   Basic Results

Our results are based on a simple 'hide-and-seek' statistical estimation problem, for which we show a strong gap between the performance of information-constrained protocols and constraint-free protocols. It is parameterized by a dimension $d$, bias $\rho$, and sample size $mn$, and defined as follows:

**Definition 2** (Hide-and-seek Problem). *Consider the set of product distributions $\{\Pr_j(\cdot)\}_{j=1}^d$ over $\{-1, 1\}^d$ defined via $\mathbb{E}_{\mathbf{x} \sim \Pr_j(\cdot)}[x_i] = 2\rho \, \mathbf{1}_{i=j}$ for all coordinates $i = 1, \ldots d$. Given an i.i.d. sample of $mn$ instances generated from $\Pr_j(\cdot)$, where $j$ is unknown, detect $j$.*

In words, $\Pr_j(\cdot)$ corresponds to picking all coordinates other than $j$ to be $\pm 1$ uniformly at random, and independently picking coordinate $j$ to be $+1$ with a higher probability $\left(\frac{1}{2} + \rho\right)$. The goal is to detect the biased coordinate $j$ based on a sample.

First, we note that without information constraints, it is easy to detect the biased coordinate with $\mathcal{O}(\log(d)/\rho^2)$ instances. This is formalized in the following theorem, which is an immediate consequence of Hoeffding's inequality and a union bound:

**Theorem 1.** *Consider the hide-and-seek problem defined earlier. Given $mn$ samples, if $\tilde{J}$ is the coordinate with the highest empirical average, then $\Pr_j(\tilde{J} = j) \geq 1 - 2d \exp\left(-\frac{1}{2}mn\rho^2\right)$.*

We now show that for this hide-and-seek problem, there is a large regime where detecting $j$ is information-theoretically possible (by Thm. 1), but any information-constrained protocol will fail to do so with high probability. We first show this for $(b, 1, m)$ protocols (i.e. protocols which process one instance at a time, such as bounded-memory online algorithms, and distributed algorithms where each machine holds a single instance):

**Theorem 2.** *Consider the hide-and-seek problem on $d > 1$ coordinates, with some bias $\rho \leq 1/4$ and sample size $m$. Then for any estimate $\tilde{J}$ of the biased coordinate returned by any $(b, 1, m)$ protocol, there exists some coordinate $j$ such that*

$$\Pr_j(\tilde{J} = j) \leq \frac{3}{d} + 21\sqrt{m\frac{\rho^2 b}{d}}.$$

The theorem implies that any algorithm corresponding to $(b, 1, m)$ protocols requires sample size $m \geq \Omega((d/b)/\rho^2)$ to reliably detect some $j$. When $b$ is polynomially smaller than $d$ (e.g. a constant), we get an exponential gap compared to constraint-free protocols, which only require $\mathcal{O}(\log(d)/\rho^2)$ instances.

Moreover, Thm. 2 is tight up to log-factors: Consider a $b$-memory online algorithm, which splits the $d$ coordinates into $\mathcal{O}(d/b)$ segments of $\mathcal{O}(b)$ coordinates each, and sequentially goes over the segments, each time using $\tilde{\mathcal{O}}(1/\rho^2)$ independent instances to determine if one of the coordinates in each segment is biased by $\rho$ (assuming $\rho$ is not exponentially smaller than $b$, this can be done with $\mathcal{O}(b)$ memory by maintaining the empirical average of each coordinate). This will allow to detect the biased coordinate, using $\tilde{\mathcal{O}}((d/b)/\rho^2)$ instances.

We now turn to provide an analogous result for general $(b, n, m)$ protocols (where $n$ is possibly greater than 1). However, it is a bit weaker in terms of the dependence on the bias parameter[1]:

**Theorem 3.** *Consider the hide-and-seek problem on $d > 1$ coordinates, with some bias $\rho \leq 1/4n$ and sample size $mn$. Then for any estimate $\tilde{J}$ of the biased coordinate returned by any $(b, n, m)$ protocol, there exists some coordinate $j$ such that*

$$\mathrm{Pr}_j(\tilde{J} = j) \leq \frac{3}{d} + 5\sqrt{mn \min\left\{\frac{10\rho b}{d}, \rho^2\right\}}.$$

The theorem implies that any $(b, n, m)$ protocol will require a sample size $mn$ which is at least $\Omega\left(\max\left\{\frac{(d/b)}{\rho}, \frac{1}{\rho^2}\right\}\right)$ in order to detect the biased coordinate. This is larger than the $\mathcal{O}(\log(d)/\rho^2)$ instances required by constraint-free protocols whenever $\rho > b\log(d)/d$, and establishes trade-offs between sample complexity and information complexities such as memory and communication.

Due to lack of space, all our proofs appear in the supplementary material. However, the technical details may obfuscate the high-level intuition, which we now turn to explain.

From an information-theoretic viewpoint, our results are based on analyzing the mutual information between $j$ and $W^t$ in a graphical model as illustrated in figure 1. In this model, the unknown message $j$ (i.e. the identity of the biased coordinate) is correlated with one of $d$ independent binary-valued random vectors (one for each coordinate across the data instances $X^t$). All these random vectors are noisy, and the mutual information in bits between $X_j^t$ and $j$ can be shown to be on the order of $n\rho^2$. Without information constraints, it follows that given $m$ instantiations of $X^t$, the total amount of information conveyed on $j$ by the data is $\Theta(mn\rho^2)$, and if this quantity is larger than $\log(d)$, then there is enough information to uniquely identify $j$. Note that no stronger bound can be established with standard statistical lower-bound techniques, since these do not consider information constraints internal to the algorithm used.

Indeed, in our information-constrained setting there is an added complication, since the output $W^t$ can only contain $b$ bits. If $b \ll d$, then $W^t$ cannot convey all the information on $X_1^t, \ldots, X_d^t$. Moreover, it will likely convey only little information if it doesn't already "know" $j$. For example, $W^t$ may provide a little bit of information on all $d$ coordinates, but then the amount of information conveyed on each (and in particular, the random variable $X_j^t$ which is correlated with $j$) will be very small. Alternatively, $W^t$ may provide accurate information on $\mathcal{O}(b)$ coordinates, but since the relevant coordinate $j$ is not known, it is likely to "miss" it. The proof therefore relies on the following components:

- No matter what, a $(b, n, m)$ protocol cannot provide more than $b/d$ bits of information (in expectation) on $X_j^t$, unless it already "knows" $j$.

- Even if the mutual information between $W^t$ and $X_j^t$ is only $b/d$, and the mutual information between $X_j^t$ and $j$ is $n\rho^2$, standard information-theoretic tools such as the data processing inequality only implies that the mutual information between $W^t$ and $j$ is bounded by $\min\{n\rho^2, b/d\}$. We essentially prove a stronger information contraction bound, which is the *product* of the two terms

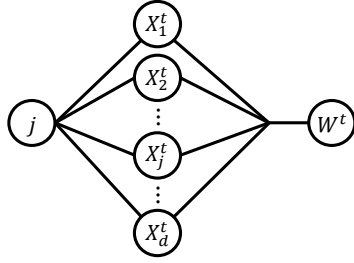

Figure 1: Illustration of the relationship between $j$, the coordinates $1, 2, \ldots, j, \ldots, d$ of the sample $X^t$, and the message $W^t$. The coordinates are independent of each other, and most of them just output $\pm 1$ uniformly at random. Only $X_j^t$ has a slightly different distribution and hence contains some information on $j$.

$\mathcal{O}(\rho^2 b/d)$ when $n = 1$, and $\mathcal{O}(n\rho b/d)$ for general $n$. At a technical level, this is achieved by considering the relative entropy between the distributions of $W^t$ with and without a biased coordinate $j$, relating it to the $\chi^2$-divergence between these distributions (using relatively recent analytic results on Csiszár f-divergences [16], [26]), and performing algebraic manipulations to upper bound it by $\rho^2$ times the mutual information between $W^t$ and $X_j^t$, which is on average $b/d$ as discussed earlier. This eventually leads to the $m\rho^2 b/d$ term in Thm. 2, as well as Thm. 3 using somewhat different calculations.

## 4 Applications

### 4.1 Online Learning with Partial Information

Consider the setting of learning with expert advice, defined as a game over $T$ rounds, where each round $t$ a loss vector $\ell_t \in [0, 1]^d$ is chosen, and the learner (without knowing $\ell_t$) needs to pick an action $i_t$ from a fixed set $\{1, \ldots, d\}$, after which the learner suffers loss $\ell_{t,i_t}$. The goal of the learner is to minimize the regret with respect to any fixed action $i$, $\sum_{t=1}^T \ell_{t,i_t} - \sum_{t=1}^T \ell_{t,i}$. We are interested in variants where the learner only gets some partial information on $\ell_t$. For example, in multi-armed bandits, the learner can only view $\ell_{t,i_t}$. The following theorem is a simple corollary of Thm. 2:

**Theorem 4.** *Suppose $d > 3$. For any $(b, 1, T)$ protocol, there is an i.i.d. distribution over loss vectors $\ell_t \in [0, 1]^d$ for which $\min_j \mathbb{E}\left[\sum_{t=1}^T \ell_{t,j_t} - \sum_{t=1}^T \ell_{t,j}\right] \geq c \, \min\left\{T, \sqrt{(d/b)/T}\right\}$, where $c > 0$ is a numerical constant.*

As a result, we get that for any algorithm with any partial information feedback model (where $b$ bits are extracted from each $d$-dimensional loss vector), it is impossible to get regret lower than $\Omega(\sqrt{(d/b)T})$ for sufficiently large $T$. Without further assumptions on the feedback model, the bound is optimal up to log-factors, as shown by $\mathcal{O}(\sqrt{(d/b)T})$ upper bounds for linear or coordinate measurements (where $b$ is the number of measurements or coordinates seen[2]) [2, 19, 25]. However, the lower bound extends beyond these specific settings, and include cases such as arbitrary nonlinear measurements of the loss vector, or receiving feedback signals of bounded size (although some setting-specific lower bounds may be stronger). It also simplifies previous lower bounds, tailored to specific types of partial information feedback, or relying on careful reductions to multi-armed bandits (e.g. [12, 25]). Interestingly, the bound holds even if the algorithm is allowed to examine each loss vector $\ell_t$ and adaptively choose which $b$ bits of information it wishes to retain.

### 4.2 Stochastic Optimization

We now turn to consider an example from stochastic optimization, where our goal is to approximately minimize $F(\mathbf{h}) = \mathbb{E}_Z[f(\mathbf{h}; Z)]$ given access to $m$ i.i.d. instantiations of $Z$, whose distribution is unknown. This setting has received much attention in recent years, and can be used to model many statistical learning problems. In this section, we demonstrate a stochastic optimization problem where information-constrained protocols provably pay a performance price compared to non-constrained algorithms. We emphasize that it is a simple toy problem, and not meant to represent anything realistic. We present it for two reasons: First, it illustrates another type of situation

where information-constrained protocols may fail (in particular, problems involving matrices). Second, the intuition of the construction is also used in the more realistic problem of sparse PCA and covariance estimation, considered in the next section.

Specifically, suppose we wish to solve $\min_{(\mathbf{w},\mathbf{v})} F(\mathbf{w},\mathbf{v}) = \mathbb{E}_Z[f((\mathbf{w},\mathbf{v});Z)]$, where

$$f((\mathbf{w},\mathbf{v});Z) = \mathbf{w}^\top Z \mathbf{v} \ , \ \ Z \in [-1,+1]^{d \times d}$$

and $\mathbf{w}, \mathbf{v}$ range over all vectors in the simplex (i.e. $w_i, v_i \geq 0$ and $\sum_{i=1}^d w_i = \sum_{i=1}^d v_i = 1$). A minimizer of $F(\mathbf{w},\mathbf{v})$ is $(\mathbf{e}_{i^*}, \mathbf{e}_{j^*})$, where $(i^*, j^*)$ are indices of the matrix entry with minimal mean. Moreover, by a standard concentration of measure argument, given $m$ i.i.d. instantiations $Z^1, \ldots, Z^m$ from any distribution over $Z$, then the solution $(\mathbf{e}_{\tilde{I}}, \mathbf{e}_{\tilde{J}})$, where $(\tilde{I}, \tilde{J}) = \arg\min_{i,j} \frac{1}{m}\sum_{t=1}^m Z_{i,j}^t$ are the indices of the entry with empirically smallest mean, satisfies $F(\mathbf{e}_{\tilde{I}}, \mathbf{e}_{\tilde{J}}) \leq \min_{\mathbf{w},\mathbf{v}} F(\mathbf{w},\mathbf{v}) + \mathcal{O}\left(\sqrt{\log(d)/m}\right)$ with high probability.

However, computing $(\tilde{I}, \tilde{J})$ as above requires us to track $d^2$ empirical means, which may be expensive when $d$ is large. If instead we constrain ourselves to $(b, 1, m)$ protocols where $b = \mathcal{O}(d)$ (e.g. any sort of stochastic gradient method optimization algorithm, whose memory is linear in the number of parameters), then we claim that we have a lower bound of $\Omega(\min\{1, \sqrt{d/m}\})$ on the expected error, which is much higher than the $\mathcal{O}(\sqrt{\log(d)/m})$ upper bound for constraint-free protocols. This claim is a straightforward consequence of Thm. 2: We consider distributions where $Z \in \{-1,+1\}^{d \times d}$ with probability 1, each of the $d^2$ entries is chosen independently, and $\mathbb{E}[Z]$ is zero except some coordinate $(i^*, j^*)$ where it equals $\mathcal{O}(\sqrt{d/m})$. For such distributions, getting optimization error smaller than $\mathcal{O}(\sqrt{d/m})$ reduces to detecting $(i^*, j^*)$, and this in turn reduces to the hide-and-seek problem defined earlier, over $d^2$ coordinates and a bias $\rho = \mathcal{O}(\sqrt{d/m})$. However, Thm. 2 shows that no $(b, 1, m)$ protocol (where $b = \mathcal{O}(d)$) will succeed if $md\rho^2 \ll d^2$, which indeed happens if $\rho$ is small enough.

Similar kind of gaps can be shown using Thm. 3 for general $(b, n, m)$ protocols, which apply to any special case such as non-interactive distributed learning.

### 4.3 Sparse PCA, Sparse Covariance Estimation, and Detecting Correlations

The sparse PCA problem ([31]) is a standard and well-known statistical estimation problem, defined as follows: We are given an i.i.d. sample of vectors $\mathbf{x} \in \mathbb{R}^d$, and we assume that there is some direction, corresponding to some *sparse* vector $\mathbf{v}$ (of cardinality at most $k$), such that the variance $\mathbb{E}[(\mathbf{v}^\top \mathbf{x})^2]$ along that direction is larger than at any other direction. Our goal is to find that direction.

We will focus here on the simplest possible form of this problem, where the maximizing direction $\mathbf{v}$ is assumed to be 2-sparse, i.e. there are only 2 non-zero coordinates $v_i, v_j$. In that case, $\mathbb{E}[(\mathbf{v}^\top \mathbf{x})^2] = v_1^2 \mathbb{E}[x_1^2] + v_2^2 \mathbb{E}[x_2^2] + 2v_1 v_2 \mathbb{E}[\mathbf{x}_i \mathbf{x}_j]$. Following previous work (e.g. [8]), we even assume that $\mathbb{E}[x_i^2] = 1$ for all $i$, in which case the sparse PCA problem reduces to detecting a coordinate pair $(i^*, j^*), i^* < j^*$ for which $x_{i^*}, x_{j^*}$ are maximally correlated. A special case is a simple and natural sparse covariance estimation problem [9], where we assume that all covariates are uncorrelated ($\mathbb{E}[x_i x_j] = 0$) except for a unique correlated pair $(i^*, j^*)$ which we need to detect.

This setting bears a resemblance to the example seen in the context of stochastic optimization in section 4.2: We have a $d \times d$ stochastic matrix $\mathbf{x}\mathbf{x}^\top$, and we need to detect an off-diagonal biased entry at location $(i^*, j^*)$. Unfortunately, these stochastic matrices are rank-1, and do not have independent entries as in the example considered in section 4.2. Instead, we use a more delicate construction, relying on distributions supported on sparse vectors. The intuition is that then each instantiation of $\mathbf{x}\mathbf{x}^\top$ is sparse, and the situation can be reduced to a variant of our hide-and-seek problem where only a few coordinates are non-zero at a time. The theorem below establishes performance gaps between constraint-free protocols (in particular, a simple plug-in estimator), and any $(b, n, m)$ protocol for a specific choice of $n$, or any $b$-memory online protocol (See Sec. 2).

**Theorem 5.** *Consider the class of* 2*-sparse PCA (or covariance estimation) problems in* $d \geq 9$ *dimensions as described above, and all distributions such that* $\mathbb{E}[x_i^2] = 1$ *for all* $i$*, and:*

1. *For a unique pair of distinct coordinates* $(i^*, j^*)$*, it holds that* $\mathbb{E}[x_{i^*} x_{j^*}] = \tau > 0$*, whereas* $\mathbb{E}[x_i x_j] = 0$ *for all distinct coordinate pairs* $(i, j) \neq (i^*, j^*)$*.*

2. *For any $i < j$, if $\widetilde{x_i x_j}$ is the empirical average of $x_i x_j$ over $m$ i.i.d. instances, then*
$$\Pr\left(|\widetilde{x_i x_j} - \mathbb{E}[x_i x_j]| \geq \tfrac{\tau}{2}\right) \leq 2\exp\left(-m\tau^2/6\right).$$

*Then the following holds:*

- *Let $(\tilde{I}, \tilde{J}) = \arg\max_{i<j} \widetilde{x_i x_j}$. Then for any distribution as above, $\Pr((\tilde{I}, \tilde{J}) = (i^*, j^*)) \geq 1 - d^2 \exp(-m\tau^2/6)$. In particular, when the bias $\tau$ equals $\Theta(1/d\log(d))$,*
$$\Pr((\tilde{I}, \tilde{J}) = (i^*, j^*)) \geq 1 - d^2 \exp\left(-\Omega\left(\frac{m}{d^2\log^2(d)}\right)\right).$$

- *For any estimate $(\tilde{I}, \tilde{J})$ of $(i^*, j^*)$ returned by any $b$-memory online protocol using $m$ instances, or any $(b, d(d-1), \lfloor\frac{m}{d(d-1)}\rfloor)$ protocol, there exists a distribution with bias $\tau = \Theta(1/d\log(d))$ as above such that*
$$\Pr\left((\tilde{I}, \tilde{J}) = (i^*, j^*)\right) \leq \mathcal{O}\left(\frac{1}{d^2} + \sqrt{\frac{m}{d^4/b}}\right).$$

The theorem implies that in the regime where $b \ll d^2/\log^2(d)$, we can choose any $m$ such that $\frac{d^4}{b} \gg m \gg d^2\log^2(d)$, and get that the chances of the protocol detecting $(i^*, j^*)$ are arbitrarily small, even though the empirical average reveals $(i^*, j^*)$ with arbitrarily high probability. Thus, in this sparse PCA / covariance estimation setting, any online algorithm with sub-quadratic memory cannot be statistically optimal for all sample sizes. The same holds for any $(b, n, m)$ protocol in an appropriate regime of $(n, m)$, such as distributed algorithms as discussed earlier.

To the best of our knowledge, this is the first result which explicitly shows that *memory* constraints can incur a statistical cost for a standard estimation problem. It is interesting that sparse PCA was also shown recently to be affected by *computational* constraints on the algorithm's runtime ([8]).

The proof appears in the supplementary material. Besides using a somewhat different hide-and-seek construction as mentioned earlier, it also relies on the simple but powerful observation that any $b$-memory online protocol is also a $(b, \kappa, \lfloor m/\kappa \rfloor)$ protocol for arbitrary $\kappa$. Therefore, we only need to prove the theorem for $(b, \kappa, \lfloor m/\kappa \rfloor)$ for some $\kappa$ (chosen to equal $d(d-1)$ in our case) to automatically get the same result for $b$-memory protocols.

## 5 Discussion and Open Questions

In this paper, we investigated cases where a generic type of information-constrained algorithm has strictly inferior statistical performance compared to constraint-free algorithms. As special cases, we demonstrated such gaps for memory-constrained and communication-constrained algorithms (e.g. in the context of sparse PCA and covariance estimation), as well as online learning with partial information and stochastic optimization. These results are based on explicitly considering the information-theoretic structure of the problem, and depend only on the number of bits extracted from each data batch.

Several questions remain open. One question is whether Thm. 3 can be improved. We conjecture this is true, and that the bound should actually depend on $mn\rho^2 b/d$ rather than $mn\min\{\rho b/d, \rho^2\}$. This would allow, for instance, to show the same type of performance gaps for $(b, 1, m)$ protocols and $(b, n, m)$ protocols. A second open question is whether there are convex stochastic optimization problems, for which online or distributed algorithms are provably inferior to constraint-free algorithms (the example discussed in section 4.2 refers to an easily-solvable yet non-convex problem). A third open question is whether our results for distributed algorithms can be extended to more interactive protocols, where the different machines can communicate over several rounds. There is a rich literature on the subject within theoretical computer science, but it is not clear how to 'import' these results to a statistical setting based on i.i.d. data. A fourth open question is whether the performance gap that we demonstrated for sparse-PCA / covariance estimation can be extended to a 'natural' distribution (e.g. Gaussian), as our result uses a tailored distribution, which has a sufficiently controlled tail behavior but is 'spiky' and not sub-Gaussian uniformly in the dimension. More generally, it would be interesting to extend the results to other learning problems and information constraints.

**Acknowledgements:** This research is supported by the Intel ICRI-CI Institute, Israel Science Foundation grant 425/13, and an FP7 Marie Curie CIG grant. We thank John Duchi, Yevgeny Seldin and Yuchen Zhang for helpful comments.

## Footnotes

[1] The proof of Thm. 2 also applies in the case $n > 1$, but the dependence on $n$ is exponential - see the proof for details.

[2]Strictly speaking, if the losses are continuous-valued, these require arbitrary-precision measurements, but in any practical implementation we can assume the losses and measurements are discrete.

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
