[Supplementary Material]

# A  Proofs

The proofs use several standard quantities and results from information theory – see Appendix B for more details. They also make use of a several auxiliary lemmas (presented in Subsection A.1), including a simple but key lemma (Lemma 6) which quantifies how information-constrained protocols cannot provide information on all coordinates simultaneously.

## A.1  Auxiliary Lemmas

**Lemma 1.** *Suppose that $d > 1$, and for some fixed distribution $\mathrm{Pr}_0(\cdot)$ over the messages $w^1, \ldots, w^m$ computed by an information-constrained protocol, it holds that*

$$\sqrt{\frac{2}{d}\sum_{j=1}^{d} D_{kl}\left(\mathrm{Pr}_0(w^1 \ldots w^m) || \mathrm{Pr}_j(w^1 \ldots w^m)\right)} \leq B.$$

*Then there exist some $j$ such that*

$$\mathrm{Pr}(\tilde{J} = j) \leq \frac{3}{d} + 2B.$$

*Proof.* By concavity of the square root, we have

$$\sqrt{\frac{2}{d}\sum_{j=1}^{d} D_{kl}\left(\mathrm{Pr}_0(w^1 \ldots w^m) || \mathrm{Pr}_j(w^1 \ldots w^m)\right)} \geq \frac{1}{d}\sum_{j=1}^{d}\sqrt{2\, D_{kl}\left(\mathrm{Pr}_0(w^1 \ldots w^m) || \mathrm{Pr}_j(w^1 \ldots w^m)\right)}.$$

Using Pinsker's inequality and the fact that $\tilde{J}$ is some function of the messages $w^1, \ldots, w^m$ (independent of the data distribution), this is at least

$$\frac{1}{d}\sum_{j=1}^{d}\sum_{w^1 \ldots w^m}\left|\mathrm{Pr}_0(w^1 \ldots w^m) - \mathrm{Pr}_j(w^1 \ldots w^m)\right|$$

$$\geq \frac{1}{d}\sum_{j=1}^{d}\left|\sum_{w^1 \ldots w^m}\left(\mathrm{Pr}_0(w^1 \ldots w^m) - \mathrm{Pr}_j(w^1 \ldots w^m)\right)\mathrm{Pr}\left(\tilde{J}|w^1 \ldots w^m\right)\right|$$

$$\geq \frac{1}{d}\sum_{j=1}^{d}\left|\mathrm{Pr}_0(\tilde{J} = j) - \mathrm{Pr}_j(\tilde{J} = j)\right|.$$

Thus, we may assume that

$$\frac{1}{d}\sum_{j=1}^{d}\left|\mathrm{Pr}_0(\tilde{J} = j) - \mathrm{Pr}_j(\tilde{J} = j)\right| \leq B.$$

The argument now uses a basic variant of the probabilistic method. If the expression above is at most $B$, then for at least $d/2$ values of $j$, it holds that $|\mathrm{Pr}_0(\tilde{J} = j) - \mathrm{Pr}_j(\tilde{J} = j)| \leq 2B$. Also, since $\sum_{j=1}^{d} \mathrm{Pr}_0(\tilde{J} = j) = 1$, then for at least $2d/3$ values of $j$, it holds that $\mathrm{Pr}_0(\tilde{J} = j) \leq 3/d$. Combining the two observations, and assuming that $d > 1$, it means there must exist some value of $j$ such that $|\mathrm{Pr}_0(\tilde{J}) - \mathrm{Pr}_j(\tilde{J} = j)| \leq 2B$, as well as $\mathrm{Pr}_0(\tilde{J} = j) \leq 3/d$, hence $\mathrm{Pr}_j(\tilde{J} = j) \leq \frac{3}{d} + 2B$ as required. $\qquad\square$

**Lemma 2.** *Let $p, q$ be distributions over a product domain $A_1 \times A_2 \times \ldots \times A_d$, where each $A_i$ is a finite set. Suppose that for some $j \in \{1, \ldots, d\}$, the following inequality holds for all $\mathbf{z} = (z_1, \ldots, z_d) \in A_1 \times \ldots \times A_d$:*

$$p(\{z_i\}_{i \neq j}|z_j) = q(\{z_i\}_{i \neq j}|z_j).$$

*Also, let $E$ be an event such that $p(E|\mathbf{z}) = q(E|\mathbf{z})$ for all $\mathbf{z}$. Then*

$$p(E) = \sum_{z_j} p(z_j)q(E|z_j).$$

*Proof.*

$$p(E) = \sum_{\mathbf{z}} p(\mathbf{z})p(E|\mathbf{z}) = \sum_{\mathbf{z}} p(\mathbf{z})q(E|\mathbf{z})$$

$$= \sum_{z_j} p(z_j) \sum_{\{z_i\}_{i \neq j}} p(\{z_j\}_{i \neq j}|z_j)q(E|z_j, \{z_i\}_{i \neq j})$$

$$= \sum_{z_j} p(z_j) \sum_{\{z_i\}_{i \neq j}} q(\{z_j\}_{i \neq j}|z_j)q(E|z_j, \{z_i\}_{i \neq j})$$

$$= \sum_{z_j} p(z_j)q(E|z_j).$$

$\square$

**Lemma 3** ([16], Proposition 1). *Let $p, q$ be two distributions on a discrete set, such that $\max_x \frac{p(x)}{q(x)} \leq c$. Then*

$$D_{kl}\left(p(\cdot)||q(\cdot)\right) \leq c \, D_{kl}\left(q(\cdot)||p(\cdot)\right).$$

**Lemma 4** ([16], Proposition 2 and Remark 4). *Let $p, q$ be two distributions on a discrete set, such that $\max_x \frac{p(x)}{q(x)} \leq c$. Also, let $D_{\chi^2}(p(\cdot)||q(\cdot)) = \sum_x \frac{(p(x)-q(x))^2}{q(x)}$ denote the $\chi^2$-divergence between the distributions $p, q$. Then*

$$D_{kl}\left(p(\cdot)||q(\cdot)\right) \leq D_{\chi^2}\left(p(\cdot)||q(\cdot)\right) \leq 2c \, D_{kl}\left(p(\cdot)||q(\cdot)\right).$$

**Lemma 5.** *Suppose we throw $n$ balls independently and uniformly at random into $d > 1$ bins, and let $K_1, \ldots K_d$ denote the number of balls in each of the $d$ bins. Then for any $\epsilon \geq 0$ such that $\epsilon \leq \min\{\frac{1}{6}, \frac{1}{2\log(d)}, \frac{d}{3n}\}$, it holds that*

$$\mathbb{E}\left[\exp\left(\epsilon \max_j K_j\right)\right] < 13.$$

*Proof.* Each $K_j$ can be written as $\sum_{i=1}^{n} \mathbf{1}(\text{ball } i \text{ fell into bin } j)$, and has expectation $n/d$. Therefore, by a standard multiplicative Chernoff bound, for any $\gamma \geq 0$,

$$\Pr\left(K_j > (1+\gamma)\frac{n}{d}\right) \leq \exp\left(-\frac{\gamma^2}{2(1+\gamma)}\frac{n}{d}\right).$$

By a union bound, this implies that

$$\Pr\left(\max_j K_j > (1+\gamma)\frac{n}{d}\right) \leq \sum_{j=1}^{d} \Pr\left(K_j > (1+\gamma)\frac{n}{d}\right) \leq d\exp\left(-\frac{\gamma^2}{2(1+\gamma)}\frac{n}{d}\right).$$

In particular, if $\gamma + 1 \geq 6$, we can upper bound the above by the simpler expression $\exp(-(1+\gamma)n/3d)$. Letting $\tau = \gamma + 1$, we get that for any $\tau \geq 6$,

$$\Pr\left(\max_j K_j > \tau\frac{n}{d}\right) \leq d\exp\left(-\frac{\tau n}{3d}\right). \tag{1}$$

Define $c = \max\{8, d^{3\epsilon}\}$. Using the inequality above and the non-negativity of $\exp(\epsilon \max_j K_j)$, we have

$$\mathbb{E}\left[\exp(\epsilon \max_j K_j)\right] = \int_{t=0}^{\infty} \Pr\left(\exp(\epsilon \max_j K_j) \geq t\right) dt$$

$$\leq c + \int_{t=c}^{\infty} \Pr\left(\exp(\epsilon \max_j K_j) \geq t\right) dt$$

$$= c + \int_{t=c}^{\infty} \Pr\left(\max_j K_j \geq \frac{\log(t)}{\epsilon}\right) dt$$

$$= c + \int_{t=c}^{\infty} \Pr\left(\max_j K_j \geq \frac{\log(t)d}{\epsilon n}\frac{n}{d}\right) dt$$

Since we assume $\epsilon \leq d/3n$ and $c \geq 8$, it holds that $\exp(6\epsilon n/d) \leq \exp(2) < 8 \leq c$, which implies $\log(c)d/\epsilon n \geq 6$. Therefore, for any $t \geq c$, it holds that $\log(t)d/\epsilon n \geq 6$. This allows us to use Eq. (1) to upper bound the expression above by

$$c + d \int_{t=c}^{\infty} \exp\left(-\frac{\log(t)d}{3\epsilon n}\frac{n}{d}\right) dt = c + d \int_{t=c}^{\infty} t^{-1/3\epsilon} dt.$$

Since we assume $\epsilon \leq 1/6$, we have $1/(3\epsilon) \geq 2$, and therefore we can solve the integration to get

$$c + \frac{d}{\frac{1}{3\epsilon}-1}c^{1-\frac{1}{3\epsilon}} \leq c + dc^{1-\frac{1}{3\epsilon}}.$$

Using the value of $c$, and since $1 - \frac{1}{3\epsilon} \leq -1$, this is at most

$$\max\{8, d^{3\epsilon}\} + d * \left(d^{3\epsilon}\right)^{1-\frac{1}{3\epsilon}} = \max\{8, d^{3\epsilon}\} + d^{3\epsilon}.$$

Since $\epsilon \leq 1/2\log(d)$, this is at most

$$\max\{8, \exp(3/2)\} + \exp(3/2) < 13$$

as required. $\qquad\square$

**Lemma 6.** *Let $Z_1, \ldots, Z_d$ be independent random variables, and let $W$ be a random variable which can take at most $2^b$ values. Then*

$$\frac{1}{d}\sum_{j=1}^{d} I(W; Z_j) \leq \frac{b}{d}.$$

*Proof.* We have

$$\frac{1}{d}\sum_{j=1}^{d} I(W; Z_j) = \frac{1}{d}\sum_{j=1}^{d}\left(H(Z_j) - H(Z_j|W)\right).$$

Using the fact that $\sum_{j=1}^{d} H(Z_j|W) \geq H(Z_1 \ldots, Z_d|W)$, this is at most

$$\frac{1}{d}\sum_{j=1}^{d} H(Z_j) - \frac{1}{d}H(Z_1 \ldots Z_d|W)$$

$$= \frac{1}{d}\sum_{j=1}^{d} H(Z_j) - \frac{1}{d}\left(H(Z_1 \ldots Z_d) - I(Z_1 \ldots Z_d; W)\right)$$

$$= \frac{1}{d}I(Z_1 \ldots Z_d; W) + \frac{1}{d}\left(\sum_{j=1}^{d} H(Z_j) - H(Z_1 \ldots Z_d)\right). \qquad (2)$$

Since $Z_1 \ldots Z_d$ are independent, $\sum_{j=1}^{d} H(Z_j) = H(Z_1 \ldots Z_d)$, hence the above equals

$$\frac{1}{d}I(Z_1 \ldots Z_d; W) = \frac{1}{d}\left(H(W) - H(W|Z_1 \ldots Z_d)\right) \leq \frac{1}{d}H(W),$$

which is at most $b/d$ since $W$ is only allowed to have $2^b$ values. $\qquad\square$

## A.2 Proof of Thm. 2

We will actually prove a more general result, stating that for any $(b, n, m)$ protocol,

$$\Pr_j(\tilde{J} = j) \leq \frac{3}{d} + 14.3\sqrt{mn2^n\frac{\rho^2 b}{d}}.$$

The result stated in the theorem follows in the case $n = 1$.

The proof builds on the auxiliary lemmas presented in Appendix A.1.

On top of the distributions $\Pr_j(\cdot)$ defined in the hide-and-seek problem (Definition 2), we define an additional 'reference' distribution $\Pr_0(\cdot)$, which corresponds to the instances $\mathbf{x}$ chosen uniformly at random from $\{-1, +1\}^d$ (i.e. there is no biased coordinate).

Let $w^1, \ldots, w^m$ denote the messages computed by the protocol. It is enough to prove that

$$\frac{2}{d} \sum_{j=1}^{d} D_{kl} \left( \Pr_0(w^1 \ldots w^m) || \Pr_j(w^1 \ldots w^m) \right) \leq 51 mn2^n \rho^2 b/d, \tag{3}$$

since then by applying Lemma 1, we get that for some $j$, $\Pr_j(\tilde{J} = j) \leq (3/d) + 2\sqrt{51mn2^n\rho^2 b/d} \leq (3/d) + 14.3\sqrt{mn2^n\rho^2 b/d}$ as required.

Using the chain rule, the left hand side in Eq. (3) equals

$$\frac{2}{d} \sum_{j=1}^{d} \sum_{t=1}^{m} \mathbb{E}_{w^1 \ldots w^{t-1} \sim \Pr_0} \left[ D_{kl} \left( \Pr_0(w^t | w^1 \ldots w^{t-1}) || \Pr_j(w^t | w^1 \ldots w^{t-1}) \right) \right]$$

$$= 2 \sum_{t=1}^{m} \mathbb{E}_{w^1 \ldots w^{t-1} \sim \Pr_0} \left[ \frac{1}{d} \sum_{j=1}^{d} D_{kl} \left( \Pr_0(w^t | w^1 \ldots w^{t-1}) || \Pr_j(w^t | w^1 \ldots w^{t-1}) \right) \right] \tag{4}$$

Let us focus on a particular choice of $t$ and values $w^1 \ldots w^{t-1}$. To simplify the presentation, we drop the $^t$ superscript from the message $w^t$, and denote the previous messages $w^1 \ldots w^{t-1}$ as $\hat{w}$. Thus, we consider the quantity

$$\frac{1}{d} \sum_{j=1}^{d} D_{kl} \left( \Pr_0(w|\hat{w}) || \Pr_j(w|\hat{w}) \right). \tag{5}$$

Recall that $w$ is some function of $\hat{w}$ and a set of $n$ independent instances received in the current round. Let $\mathbf{x}_j$ denote the vector of values at coordinate $j$ across these $n$ instances. Clearly, under $\Pr_j$, every $\mathbf{x}_i$ for $i \neq j$ is uniformly distributed on $\{-1, +1\}^n$, whereas each entry of $\mathbf{x}_j$ equals 1 with probability $\frac{1}{2} + \rho$, and $-1$ otherwise.

First, we argue that by Lemma 2, for any $w, \hat{w}$, we have

$$\Pr_j(w|\hat{w}) = \Pr_0(w|\hat{w}) \sum_{\mathbf{x}_j} \Pr_j(\mathbf{x}_j | \hat{\mathbf{w}}) = \sum_{\mathbf{x}_j} \Pr_0(w|\hat{w}) \Pr_j(\mathbf{x}_j | \hat{\mathbf{w}}) = \sum_{\mathbf{x}_j} \Pr_0(w|\hat{w}) \Pr_j(\mathbf{x}_j).$$

$$\tag{6}$$

This follows by applying the lemma on $p(\cdot) = \Pr_j(\cdot|\hat{w}), q(\cdot) = \Pr_0(\cdot|\hat{w})$ and $A_i = \{-1, +1\}^n$ (i.e. the vector of values at a single coordinate $i$ across the $n$ data points), and noting the $\mathbf{x}_j$ is independent of $\hat{w}$. The lemma's conditions are satisfied since $\mathbf{x}_i$ for $i \neq j$ has the same distribution under $\Pr_0(\cdot|\hat{w})$ and $\Pr_j(\cdot|\hat{w})$, and also $w$ is only a function of $\mathbf{x}_1 \ldots \mathbf{x}_d$ and $\hat{w}$.

Using Lemma 3 and Lemma 4, we have the following.

$$D_{kl} \left( \Pr_0(w|\hat{w}) || \Pr_j(w|\hat{w}) \right) \leq \max_w \left( \frac{\Pr_0(w|\hat{w})}{\Pr_j(w|\hat{w})} \right) D_{kl} \left( \Pr_j(w|\hat{w}) || \Pr_0(w|\hat{w}) \right)$$

$$\leq \max_w \left( \frac{\Pr_0(w|\hat{w})}{\Pr_j(w|\hat{w})} \right) D_{\chi^2} \left( \Pr_j(w|\hat{w}) || \Pr_0(w|\hat{w}) \right)$$

$$= \max_w \left( \frac{\Pr_0(w|\hat{w})}{\Pr_j(w|\hat{w})} \right) \sum_w \frac{\left( \Pr_j(w|\hat{w}) - \Pr_0(w|\hat{w}) \right)^2}{\Pr_0(w|\hat{w})} \tag{7}$$

Let us consider the max term and the sum seperately. Using Eq. (6) and the fact that $\rho \leq 1/4n$, we have

$$\max_w \left( \frac{\Pr_0(w|\hat{w})}{\Pr_j(w|\hat{w})} \right) = \max_w \left( \frac{\sum_{\mathbf{x}_j} \Pr_0(w|\mathbf{x}_j, \hat{w}) \Pr_0(\mathbf{x}_j)}{\sum_{\mathbf{x}_j} \Pr_0(w|\mathbf{x}_j, \hat{w}) \Pr_j(\mathbf{x}_j)} \right)$$

$$\leq \max_{\mathbf{x}_j} \left( \frac{\Pr_0(\mathbf{x}_j)}{\Pr_j(\mathbf{x}_j)} \right)$$

$$= \left( \frac{1/2}{1/2 - \rho} \right)^n \leq (1 + 4\rho)^n \leq (1 + 1/n)^n \leq \exp(1). \tag{8}$$

As to the sum term in Eq. (7), using Eq. (6) and the Cauchy-Schwartz inequality, we have

$$
\sum_w \frac{(\mathrm{Pr}_j(w|\hat{w}) - \mathrm{Pr}_0(w|\hat{w}))^2}{\mathrm{Pr}_0(w|\hat{w})} = \sum_w \frac{\left(\sum_{\mathbf{x}_j} \mathrm{Pr}_0(w|\mathbf{x}_j, \hat{w})\,(\mathrm{Pr}_j(\mathbf{x}_j) - \mathrm{Pr}_0(\mathbf{x}_j))\right)^2}{\mathrm{Pr}_0(w|\hat{w})}
$$

$$
= \sum_w \frac{\left(\sum_{\mathbf{x}_j} (\mathrm{Pr}_0(w|\mathbf{x}_j, \hat{w}) - \mathrm{Pr}_0(w|\hat{w}))\,(\mathrm{Pr}_j(\mathbf{x}_j) - \mathrm{Pr}_0(\mathbf{x}_j))\right)^2}{\mathrm{Pr}_0(w|\hat{w})}
$$

$$
\leq \sum_w \frac{\sum_{\mathbf{x}_j} (\mathrm{Pr}_0(w|\mathbf{x}_j, \hat{w}) - \mathrm{Pr}_0(w|\hat{w}))^2 \sum_{\mathbf{x}_j} (\mathrm{Pr}_j(\mathbf{x}_j) - \mathrm{Pr}_0(\mathbf{x}_j))^2}{\mathrm{Pr}_0(w|\hat{w})}
$$

$$
= \sum_{\mathbf{x}_j} (\mathrm{Pr}_j(\mathbf{x}_j) - \mathrm{Pr}_0(\mathbf{x}_j))^2 \sum_{\mathbf{x}_j} D_{\chi^2}\left(\mathrm{Pr}_0(w|\mathbf{x}_j, \hat{w})||\mathrm{Pr}_0(w|\hat{w})\right).
$$

$$(9)$$

where we used the definition of $\chi^2$-divergence as specified in Lemma 4. Again, we will consider each sum separately. Applying Lemma 4 and Eq. (6), we have

$$
D_{\chi^2}\left(\mathrm{Pr}_0(w|\mathbf{x}_j, \hat{w})||\mathrm{Pr}_0(w|\hat{w})\right) \leq 2 \max_w \left(\frac{\mathrm{Pr}_0(w|\mathbf{x}_j, \hat{w})}{\mathrm{Pr}_0(w|\hat{w})}\right) D_{kl}\left(\mathrm{Pr}_0(w|\mathbf{x}_j, \hat{w})||\mathrm{Pr}_0(w|\hat{w})\right)
$$

$$
= 2 \max_w \left(\frac{\mathrm{Pr}_0(w|\mathbf{x}_j, \hat{w})}{\sum_{\mathbf{x}_j} \mathrm{Pr}_0(w|\mathbf{x}_j, \hat{w})\mathrm{Pr}_0(\mathbf{x}_j)}\right) D_{kl}\left(\mathrm{Pr}_0(w|\mathbf{x}_j, \hat{w})||\mathrm{Pr}_0(w|\hat{w})\right)
$$

$$
= 2 \max_w \left(\frac{\mathrm{Pr}_0(w|\mathbf{x}_j, \hat{w})}{\frac{1}{2^n} \sum_{\mathbf{x}_j} \mathrm{Pr}_0(w|\mathbf{x}_j, \hat{w})}\right) D_{kl}\left(\mathrm{Pr}_0(w|\mathbf{x}_j, \hat{w})||\mathrm{Pr}_0(w|\hat{w})\right)
$$

$$
\leq 2^{n+1} D_{kl}\left(\mathrm{Pr}_0(w|\mathbf{x}_j, \hat{w})||\mathrm{Pr}_0(w|\hat{w})\right) \qquad (10)
$$

Moreover, by definition of $\mathrm{Pr}_0$ and $\mathrm{Pr}_j$, and using the fact that each coordinate of $\mathbf{x}_j$ takes values in $\{-1, +1\}$, we have

$$
\sum_{\mathbf{x}_j} (\mathrm{Pr}_j(\mathbf{x}_j) - \mathrm{Pr}_0(\mathbf{x}_j))^2 = \sum_{\mathbf{x}_j} \left(\prod_{i=1}^n \left(\frac{1}{2} + \rho x_{j,i}\right) - \frac{1}{2^n}\right)^2
$$

$$
= \frac{1}{4^n} \sum_{\mathbf{x}_j} \left(\prod_{i=1}^n (1 + 2\rho x_{j,i}) - 1\right)^2 = \frac{1}{4^n} \sum_{\mathbf{x}_j} \left(\prod_{i=1}^n (1 + 2\rho x_{j,i})^2 - 2\prod_{i=1}^n (1 + 2\rho x_{j,i}) + 1\right)
$$

$$
= \frac{1}{4^n} \left(\prod_{i=1}^n \sum_{x_{j,i}} (1 + 2\rho x_{j,i})^2 - 2\prod_{i=1}^n \sum_{x_{j,i}} (1 + 2\rho x_{j,i}) + 2^n\right)
$$

$$
= \frac{1}{4^n} \left((2 + 8\rho^2)^n - 2^{n+1} + 2^n\right) = \frac{1}{2^n} \left((1 + 4\rho^2)^n - 1\right)
$$

$$
= \frac{1}{2^n} \left(\left(1 + \frac{4n\rho^2}{n}\right)^n - 1\right) \leq \frac{1}{2^n} \left(\exp(4n\rho^2) - 1\right) \leq \frac{4.6}{2^n} n\rho^2, \qquad (11)
$$

where in the last inequality we used the fact that $4n\rho^2 \leq 4n(1/4n)^2 \leq 0.25$, and $\exp(x) \leq 1 + 1.14x$ for any $x \in [0, 0.25]$. Plugging in Eq. (10) and Eq. (11) back into Eq. (9), we get that

$$
\sum_w \frac{(\mathrm{Pr}_j(w|\hat{w}) - \mathrm{Pr}_0(w|\hat{w}))^2}{\mathrm{Pr}_0(w|\hat{w})} \leq 9.2n\rho^2 \sum_{\mathbf{x}_j} D_{kl}\left(\mathrm{Pr}_0(w|\mathbf{x}_j, \hat{w})||\mathrm{Pr}_0(w|\hat{w})\right).
$$

Plugging this in turn, together with Eq. (8), into Eq. (7), we get overall that

$$
D_{kl}\left(\mathrm{Pr}_0(w|\hat{w})||\mathrm{Pr}_j(w|\hat{w})\right) \leq 9.2\exp(1)n\rho^2 \sum_{\mathbf{x}_j} D_{kl}\left(\mathrm{Pr}_0(w|\mathbf{x}_j, \hat{w})||\mathrm{Pr}_0(w|\hat{w})\right).
$$

This expression can be equivalently written as

$$9.2 \exp(1) n 2^n \rho^2 \sum_{\mathbf{x}_j} \frac{1}{2^n} D_{kl} \left( \Pr_0(w|\mathbf{x}_j, \hat{w}) || \Pr_0(w|\hat{w}) \right)$$

$$= 9.2 \exp(1) n 2^n \rho^2 \sum_{\mathbf{x}_j} \Pr_0(\mathbf{x}_j|\hat{w}) D_{kl} \left( \Pr_0(w|\mathbf{x}_j, \hat{w}) || \Pr_0(w|\hat{w}) \right)$$

$$= 9.2 \exp(1) n 2^n \rho^2 I_{\Pr_0(\cdot|\hat{w})}(w; \mathbf{x}_j)$$

where $I_{\Pr_0(\cdot|\hat{w})}(w; \mathbf{x}_j)$ denotes the mutual information between $w$ and $\mathbf{x}_j$, under the (uniform) distribution on $\mathbf{x}_j$ induced by $\Pr_0(\cdot|\hat{w})$. This allows us to upper bound Eq. (5) as follows:

$$\frac{1}{d} \sum_{j=1}^{d} D_{kl} \left( \Pr_0(w|\hat{w}) || \Pr_j(w|\hat{w}) \right) \leq 9.2 \exp(1) n 2^n \rho^2 \frac{1}{d} \sum_{j=1}^{d} I_{\Pr_0(\cdot|\hat{w})}(w; \mathbf{x}_j).$$

Since $\mathbf{x}_1, \ldots, \mathbf{x}_d$ are independent of each other and $w$ contains at most $b$ bits, we can use the key Lemma 6 to upper bound the above by $9.2 \exp(1) n 2^n \rho^2 b/d$.

To summarize, this expression constitutes an upper bound on Eq. (5), i.e. on any individual term inside the expectation in Eq. (4). Thus, we can upper bound Eq. (4) by $18.4 \exp(1) mn 2^n \rho^2 b/d < 51 mn 2^n \rho^2 b/d$. This shows that Eq. (3) indeed holds, which as explained earlier implies the required result.

## A.3 Proof of Thm. 3

The proof builds on the auxiliary lemmas presented in Appendix A.1. It begins similarly to the proof of Thm. 2, but soon diverges.

On top of the distributions $\Pr_j(\cdot)$ defined in the hide-and-seek problem (Definition 2), we define an additional 'reference' distribution $\Pr_0(\cdot)$, which corresponds to the instances $\mathbf{x}$ chosen uniformly at random from $\{-1, +1\}^d$ (i.e. there is no biased coordinate).

Let $w^1, \ldots, w^m$ denote the messages computed by the protocol. To show the upper bound, it is enough to prove that

$$\frac{2}{d} \sum_{j=1}^{d} D_{kl} \left( \Pr_0(w^1 \ldots w^m) || \Pr_j(w^1 \ldots w^m) \right) \leq \min \left\{ 60 \frac{mn\rho b}{d}, 6mn\rho^2 \right\} \quad (12)$$

since then by applying Lemma 1, we get that for some $j$, $\Pr_j(\tilde{J} = j) \leq (3/d) + 2\sqrt{\min\{60mn\rho b/d, 6mn\rho^2\}} \leq (3/d) + 5\sqrt{mn \min\{10\rho b/d, \rho^2\}}$ as required.

Using the chain rule, the left hand side in Eq. (12) equals

$$\frac{2}{d} \sum_{j=1}^{d} \sum_{t=1}^{m} \mathbb{E}_{w^1 \ldots w^{t-1} \sim \Pr_0} \left[ D_{kl} \left( \Pr_0(w^t|w^1 \ldots w^{t-1}) || \Pr_j(w^t|w^1 \ldots w^{t-1}) \right) \right]$$

$$= 2 \sum_{t=1}^{m} \mathbb{E}_{w^1 \ldots w^{t-1} \sim \Pr_0} \left[ \frac{1}{d} \sum_{j=1}^{d} D_{kl} \left( \Pr_0(w^t|w^1 \ldots w^{t-1}) || \Pr_j(w^t|w^1 \ldots w^{t-1}) \right) \right] \quad (13)$$

Let us focus on a particular choice of $t$ and values $w^1 \ldots w^{t-1}$. To simplify the presentation, we drop the $^t$ superscript from the message $w^t$, and denote the previous messages $w^1 \ldots w^{t-1}$ as $\hat{w}$. Thus, we consider the quantity

$$\frac{1}{d} \sum_{j=1}^{d} D_{kl} \left( \Pr_0(w|\hat{w}) || \Pr_j(w|\hat{w}) \right). \quad (14)$$

Recall that $w$ is some function of $\hat{w}$ and a set of $n$ independent instances received in the current round. Let $\mathbf{x}_j$ denote the vector of values at coordinate $j$ across these $n$ instances. Clearly, under

$\text{Pr}_j$, every $\mathbf{x}_i$ for $i \neq j$ is uniformly distributed on $\{-1, +1\}^n$, whereas each entry of $\mathbf{x}_j$ equals $1$ with probability $\frac{1}{2} + \rho$, and $-1$ otherwise.

We now show that Eq. (14) can be upper bounded in two different ways, one bound being $30n\rho b/d$ and the other being $3n\rho^2$. Combining the two, we get that

$$\frac{1}{d} \sum_{j=1}^{d} D_{kl}\left(\text{Pr}_0(w|\hat{w})||\text{Pr}_j(w|\hat{w})\right) \leq \min\left\{30\frac{n\rho b}{d}, 3n\rho^2\right\}. \tag{15}$$

Plugging this inequality back in Eq. (13), we validate Eq. (12), from which the result follows.

**The $3n\rho^2$ bound**

This bound essentially follows only from the fact that $\mathbf{x}_j$ is noisy, and not from the algorithm's information constraints, and is thus easier to obtain.

First, we have by Lemma 2 that for any $w, \hat{w}$,

$$\text{Pr}_j(w|\hat{w}) = \sum_{\mathbf{x}_j} \text{Pr}_0(w|\hat{w})\text{Pr}_j(\mathbf{x}_j|\hat{\mathbf{w}}) = \sum_{\mathbf{x}_j} \text{Pr}_0(w|\hat{w})\text{Pr}_j(\mathbf{x}_j)$$

(this is the same as Eq. (6), and the justification is the same).

Using this inequality, the definition of relative entropy, and the log-sum inequality, we have

$$\frac{1}{d} \sum_{j=1}^{d} D_{kl}\left(\text{Pr}_0(w|\hat{w})||\text{Pr}_j(w|\hat{w})\right) = \frac{1}{d} \sum_{j=1}^{d} \sum_{w} \text{Pr}_0(w|\hat{w}) \log\left(\frac{\text{Pr}_0(w|\hat{w})}{\text{Pr}_j(w|\hat{w})}\right)$$

$$= \frac{1}{d} \sum_{j=1}^{d} \sum_{w} \text{Pr}_0(w|\hat{w}) \left(\sum_{\mathbf{x}_j} \text{Pr}_0(\mathbf{x}_j)\right) \log\left(\frac{\sum_{\mathbf{x}_j} \text{Pr}_0(w|\mathbf{x}_j, \hat{w})\text{Pr}_0(\mathbf{x}_j)}{\sum_{\mathbf{x}_j} \text{Pr}_0(w|\mathbf{x}_j, \hat{w})\text{Pr}_j(\mathbf{x}_j)}\right)$$

$$\leq \frac{1}{d} \sum_{j=1}^{d} \sum_{w} \text{Pr}_0(w|\hat{w}) \sum_{\mathbf{x}_j} \text{Pr}_0(\mathbf{x}_j) \log\left(\frac{\text{Pr}_0(w|\mathbf{x}_j, \hat{w})\text{Pr}_0(\mathbf{x}_j)}{\text{Pr}_0(w|\mathbf{x}_j, \hat{w})\text{Pr}_j(\mathbf{x}_j)}\right)$$

$$= \frac{1}{d} \sum_{j=1}^{d} \sum_{w} \text{Pr}_0(w|\hat{w}) \sum_{\mathbf{x}_j} \text{Pr}_0(\mathbf{x}_j) \log\left(\frac{\text{Pr}_0(\mathbf{x}_j)}{\text{Pr}_j(\mathbf{x}_j)}\right)$$

$$= \frac{1}{d} \sum_{j=1}^{d} \sum_{\mathbf{x}_j} \text{Pr}_0(\mathbf{x}_j) \log\left(\frac{\text{Pr}_0(\mathbf{x}_j)}{\text{Pr}_j(\mathbf{x}_j)}\right)$$

$$= \frac{1}{d} \sum_{j=1}^{d} D_{kl}\left(\text{Pr}_0(\mathbf{x}_j)||\text{Pr}_j(\mathbf{x}_j)\right).$$

This relative entropy is between the distribution of $n$ independent Bernoulli trials with parameter $1/2$, and $n$ independent Bernoulli trials with parameter $1/2 + \rho$. This is easily verified to equal $n$ times the relative entropy for a single trial, which equals (by definition of relative entropy)

$$\frac{1}{2} \log\left(\frac{1/2}{1/2 - \rho}\right) + \frac{1}{2} \log\left(\frac{1/2}{1/2 + \rho}\right) = -\frac{1}{2} \log\left(1 - 4\rho^2\right) \leq 8\log(4/3)\rho^2,$$

where we used the fact that $\rho \leq 1/4n \leq 1/4$, and the inequality $-\log(1 - x) \leq 4\log(4/3)x$ for $x \in [0, 1/4]$. Overall, we get that

$$\frac{1}{d} \sum_{j=1}^{d} D_{kl}\left(\text{Pr}_0(w|\hat{w})||\text{Pr}_j(w|\hat{w})\right) \leq 8\log(4/3)n\rho^2 \leq 3n\rho^2.$$

**The $30n\rho b/d$ bound**

To prove this bound, it will be convenient for us to describe the sampling process of $\mathbf{x}_j$ in a slightly more complex way, as follows[3]:

- We let $\mathbf{v} \in \{0,1\}^n$ be an auxiliary random vector with independent entries, where each $v_i = 1$ with probability $4\rho$, and 0 otherwise.

- Under $\Pr_0$ and $\Pr_i$ for $i \neq j$, we assume that $\mathbf{x}_j$ is drawn uniformly from $\{-1,+1\}^n$ regardless of the value of $\mathbf{v}$.

- Under $\Pr_j$, we assume that each entry $x_{j,l}$ is independently sampled (in a manner depending on $\mathbf{v}$) as follows:

  - For each $l$ such that $v_l = 1$, we pick $x_{j,l}$ to be 1 with probability $3/4$, and $-1$ otherwise.

  - For each $l$ such that $v_l = 0$, we pick $x_{j,l}$ to be 1 or $-1$ with probability $1/2$.

Note that this induces the same distribution on $\mathbf{x}_j$ as before: Each individual entry $x_{j,l}$ is independent and satisfies $\Pr_j(x_{j,l} = 1) = 4\rho * \frac{3}{4} + (1 - 4\rho) * \frac{1}{2} = \frac{1}{2} + \rho$.

Having finished with these definitions, we re-write Eq. (14) as

$$\frac{1}{d} \sum_{j=1}^d D_{kl} \left( \mathbb{E}_\mathbf{v} \left[ \Pr_0(w|\mathbf{v}, \hat{w}) \right] \, || \, \mathbb{E}_\mathbf{v} \left[ \Pr_j(w|\mathbf{v}, \hat{w}) \right] \right).$$

Since the relative entropy is jointly convex in its arguments, and $\mathbf{v}$ is a fixed random variable, we have by Jensen's inequality that this is at most

$$\mathbb{E}_\mathbf{v} \left[ \frac{1}{d} \sum_{j=1}^d D_{kl} \left( \Pr_0(w|\mathbf{v}, \hat{w}) || \Pr_j(w|\mathbf{v}, \hat{w}) \right) \right].$$

Now, note that if $\mathbf{v} = \mathbf{0}$ (i.e. the zero-vector), then the distribution of $\mathbf{x}_1, \ldots, \mathbf{x}_d$ is the same under both $\Pr_0$ and any $\Pr_j$. Since $w$ is a function of $\mathbf{x}_1, \ldots, \mathbf{x}_d$, it follows that the distribution of $\mathbf{w}$ will be the same under both $\Pr_j$ and $\Pr_0$, and therefore the relative entropy terms will be zero. Hence, we can trivially re-write the above as

$$\mathbb{E}_\mathbf{v} \left[ \mathbf{1}_{\mathbf{v} \neq \mathbf{0}} \frac{1}{d} \sum_{j=1}^d D_{kl} \left( \Pr_0(w|\mathbf{v}, \hat{w}) || \Pr_j(w|\mathbf{v}, \hat{w}) \right) \right]. \tag{16}$$

where $\mathbf{1}_{\mathbf{v} \neq \mathbf{0}}$ is an indicator function.

We can now use Lemma 2, where $p(\cdot) = \Pr_j(\cdot|\mathbf{v}, \hat{w}), q(\cdot) = \Pr_0(\cdot|\mathbf{v}, \hat{w})$ and $A_i = \{-1, +1\}^n$ (i.e. the vector of values at a single coordinate $i$ across the $n$ data points). The lemma's conditions are satisfied since $\mathbf{x}_i$ for $i \neq j$ has the same distribution under $\Pr_0(\cdot|\mathbf{v}, \hat{w})$ and $\Pr_j(\cdot|\mathbf{v}, \hat{w})$, and also $w$ is only a function of $\mathbf{x}_1 \ldots \mathbf{x}_d$ and $\hat{w}$. Thus, we can rewrite Eq. (16) as

$$\mathbb{E}_\mathbf{v} \left[ \mathbf{1}_{\mathbf{v} \neq \mathbf{0}} \frac{1}{d} \sum_{j=1}^d D_{kl} \left( \Pr_0(w|\mathbf{v}, \hat{w}) \, \bigg|\bigg| \, \sum_{\mathbf{x}_j} \Pr_0(w|\mathbf{x}_j, \mathbf{v}, \hat{w}) \Pr_j(\mathbf{x}_j|\mathbf{v}, \hat{w}) \right) \right].$$

Using Lemma 3, we can reverse the expressions in the relative entropy term, and upper bound the above by

$$\mathbb{E}_\mathbf{v} \left[ \mathbf{1}_{\mathbf{v} \neq \mathbf{0}} \frac{1}{d} \sum_{j=1}^d \left( \max_w \frac{\Pr_0(w|\mathbf{v}, \hat{w})}{\sum_{\mathbf{x}_j} \Pr_0(w|\mathbf{x}_j, \mathbf{v}, \hat{w}) \Pr_j(\mathbf{x}_j|\mathbf{v}, \hat{w})} \right) D_{kl} \left( \sum_{\mathbf{x}_j} \Pr_0(w|\mathbf{x}_j, \mathbf{v}, \hat{w}) \Pr_j(\mathbf{x}_j|\mathbf{v}, \hat{w}) \, \bigg|\bigg| \, \Pr_0(w|\mathbf{v}, \hat{w}) \right) \right]. \tag{17}$$

The max term equals

$$\max_w \frac{\sum_{\mathbf{x}_j} \Pr_0(w|\mathbf{x}_j, \mathbf{v}, \hat{w}) \Pr_0(\mathbf{x}_j|\mathbf{v}, \hat{w})}{\sum_{\mathbf{x}_j} \Pr_0(w|\mathbf{x}_j, \mathbf{v}, \hat{w}) \Pr_j(\mathbf{x}_j|\mathbf{v}, \hat{w})} \leq \max_{\mathbf{x}_j} \frac{\Pr_0(\mathbf{x}_j|\mathbf{v}, \hat{w})}{\Pr_j(\mathbf{x}_j|\mathbf{v}, \hat{w})},$$

and using Jensen's inequality and the fact that relative entropy is convex in its arguments, we can upper bound the relative entropy term by

$$\sum_{\mathbf{x}_j} \Pr_j(\mathbf{x}_j|\mathbf{v}, \hat{w}) D_{kl} \left( \Pr_0(w|\mathbf{x}_j, \mathbf{v}, \hat{w}) \, || \, \Pr_0(w|\mathbf{v}, \hat{w}) \right)$$

$$\leq \left( \max_{\mathbf{x}_j} \frac{\Pr_j(\mathbf{x}_j|\mathbf{v}, \hat{w})}{\Pr_0(\mathbf{x}_j|\mathbf{v}, \hat{w})} \right) \sum_{\mathbf{x}_j} \Pr_0(\mathbf{x}_j|\mathbf{v}, \hat{w}) D_{kl} \left( \Pr_0(w|\mathbf{x}_j, \mathbf{v}, \hat{w}) \, || \, \Pr_0(w|\mathbf{v}, \hat{w}) \right).$$

The sum in the expression above equals the mutual information between the message $w$ and the coordinate vector $\mathbf{x}_j$ (seen as random variables with respect to the distribution $\mathrm{Pr}_0(\cdot|\mathbf{v}, \hat{w})$). Writing this as $I_{\mathrm{Pr}_0(\cdot|\mathbf{v},\hat{w})}(w; \mathbf{x}_j)$, we can thus upper bound Eq. (17) by

$$\mathbb{E}_{\mathbf{v}}\left[\mathbf{1}_{\mathbf{v}\neq\mathbf{0}}\frac{1}{d}\sum_{j=1}^{d}\left(\max_{\mathbf{x}_j}\frac{\mathrm{Pr}_0(\mathbf{x}_j|\mathbf{v},\hat{w})}{\mathrm{Pr}_j(\mathbf{x}_j|\mathbf{v},\hat{w})}\right)\left(\max_{\mathbf{x}_j}\frac{\mathrm{Pr}_j(\mathbf{x}_j|\mathbf{v},\hat{w})}{\mathrm{Pr}_0(\mathbf{x}_j|\mathbf{v},\hat{w})}\right)I_{\mathrm{Pr}_0(\cdot|\mathbf{v},\hat{w})}(w;\mathbf{x}_j)\right]$$

$$\leq \mathbb{E}_{\mathbf{v}}\left[\mathbf{1}_{\mathbf{v}\neq\mathbf{0}}\left(\max_{j,\mathbf{x}_j}\frac{\mathrm{Pr}_0(\mathbf{x}_j|\mathbf{v},\hat{w})}{\mathrm{Pr}_j(\mathbf{x}_j|\mathbf{v},\hat{w})}\right)\left(\max_{j,\mathbf{x}_j}\frac{\mathrm{Pr}_j(\mathbf{x}_j|\mathbf{v},\hat{w})}{\mathrm{Pr}_0(\mathbf{x}_j|\mathbf{v},\hat{w})}\right)\frac{1}{d}\sum_{j=1}^{d}I_{\mathrm{Pr}_0(\cdot|\mathbf{v},\hat{w})}(w;\mathbf{x}_j)\right].$$

Since $\{\mathbf{x}_j\}_j$ are independent of each other and $w$ contains at most $b$ bits, we can use the key Lemma 6 to upper bound the above by

$$\mathbb{E}_{\mathbf{v}}\left[\mathbf{1}_{\mathbf{v}\neq\mathbf{0}}\left(\max_{j,\mathbf{x}_j}\frac{\mathrm{Pr}_0(\mathbf{x}_j|\mathbf{v},\hat{w})}{\mathrm{Pr}_j(\mathbf{x}_j|\mathbf{v},\hat{w})}\right)\left(\max_{j,\mathbf{x}_j}\frac{\mathrm{Pr}_j(\mathbf{x}_j|\mathbf{v},\hat{w})}{\mathrm{Pr}_0(\mathbf{x}_j|\mathbf{v},\hat{w})}\right)\frac{b}{d}\right].$$

Now, recall that for any $j$, $\mathbf{x}_j$ refers to a column of $n$ independent entries, drawn independently of any previous messages $\hat{w}$, where under $\mathrm{Pr}_0$, each entry $x_{j,i}$ is chosen to be $\pm 1$ with equal probability, whereas under $\mathrm{Pr}_j$ each is chosen to be 1 with probability $\frac{3}{4}$ if $v_i = 1$, and with probability $\frac{1}{2}$ if $v_i = 0$. Therefore, letting $|\mathbf{v}|$ denote the number of non-zero entries in $\mathbf{v}$, we can upper bound the expression above by

$$\mathbb{E}_{\mathbf{v}}\left[\mathbf{1}_{\mathbf{v}\neq\mathbf{0}}\left(\frac{1/2}{1/4}\right)^{|\mathbf{v}|}\left(\frac{3/4}{1/2}\right)^{|\mathbf{v}|}\frac{b}{d}\right] = \frac{b}{d}\mathbb{E}_{\mathbf{v}}\left[\mathbf{1}_{\mathbf{v}\neq\mathbf{0}}3^{|\mathbf{v}|}\right], \tag{18}$$

To compute the expectation in closed-form, recall that each entry of $\mathbf{v}$ is picked independently to be 1 with probability $4\rho$, and 0 otherwise. Therefore,

$$\mathbb{E}_{\mathbf{v}}\left[\mathbf{1}_{\mathbf{v}\neq\mathbf{0}}3^{|\mathbf{v}|}\right] = \mathbb{E}_{\mathbf{v}}\left[3^{|\mathbf{v}|} - \mathbf{1}_{\mathbf{v}=\mathbf{0}}\right]$$

$$= \prod_{i=1}^{n}\mathbb{E}_{v_i}[3^{v_i}] - \mathrm{Pr}(\mathbf{v}=\mathbf{0})$$

$$= (\mathbb{E}_{v_1}[3^{v_1}])^n - \mathrm{Pr}(\mathbf{v}=\mathbf{0})$$

$$= (4\rho * 3 + (1-4\rho) * 1)^n - (1-4\rho)^n$$

$$= (1+8\rho)^n - (1-4\rho)^n \leq \exp(8n\rho) - (1-4n\rho),$$

where in the last inequality we used the facts that $(1+a/n)^n \leq \exp(a)$ and $(1-a)^n \geq 1-an$. Since we assume $\rho \leq 1/4n$, $8n\rho \leq 2$, so we can use the inequality $\exp(x) \leq 1 + 3.2x$, which holds for any $x \in [0,2]$, and get that the expression above is at most $(1+26n\rho) - (1-4n\rho) = 30n\rho$, and therefore Eq. (18) is at most $30n\rho b/d$. This in turn is an upper bound on Eq. (14) as required.

### A.4 Proof of Thm. 4

Let $c_1, c_2$ be positive parameters to be determined later, and assume by contradiction that our algorithm can guarantee $\mathbb{E}[\sum_{t=1}^{T}\ell_{t,i_t} - \sum_{t=1}^{T}\ell_{t,j}] < c_1\min\{T/4, \sqrt{dT/b}\}$ for any distribution and all $j$.

Consider the set of distributions $\mathrm{Pr}_j(\cdot)$ over $\{0,1\}^d$, where each coordinate is chosen independently and uniformly, except coordinate $j$ which equals 0 with probability $\frac{1}{2} + \rho$, where $\rho = c_2\min\{1/4, \sqrt{d/bT}\}$. Clearly, the coordinate $i$ which minimizes $\mathbb{E}[\ell_{t,i}]$ is $j$. Moreover, if at round $t$ the learner chooses some $i_t \neq j$, then $\mathbb{E}[\ell_{t,i_t} - \ell_{t,j}] = \rho = c_2\min\{1/4, \sqrt{d/bT}\}$. Thus, to have $\mathbb{E}[\sum_{t=1}^{T}\ell_{t,i_t} - \sum_{t=1}^{T}\ell_{t,j}] < c_1\min\{T/4, \sqrt{dT/b}\}$ requires that the expected number of rounds where $i_t \neq j$ is at most $\frac{c_1}{c_2}T$. By Markov's inequality, this means that the probability of $j$ not being the most-commonly chosen coordinate is at most $(c_1/c_2)/(1/2) = 2c_1/c_2$. In other words, if we can guarantee regret smaller than $c_1\min\{T/4, \sqrt{dT/b}\}$, then we can detect $j$ with probability at least $1 - 2c_1/c_2$, simply by taking the most common coordinate.

However, by[4] Thm. 2, for any $(b, 1, T)$ protocol, there is some $j$ such that the protocol would correctly detect $j$ with probability at most

$$\frac{3}{d} + 21\sqrt{\frac{Tb}{d}c_2^2 \min\left\{\frac{1}{16}, \frac{d}{bT}\right\}} \leq \frac{3}{d} + 21c_2.$$

Therefore, assuming $d > 3$, and taking for instance $c_1 = 3.7 * 10^{-4}, c_2 = 5.9 * 10^{-3}$, we get that the probability of detection is at most $\frac{3}{4} + 21c_2 < 0.874$, whereas the scheme discussed in the previous paragraph guarantees detection with probability at least $1 - 2c_1/c_2 > 0.874$. We have reached a contradiction, hence our initial hypothesis is false and our algorithm must suffer regret at least $c_1 \min\{T/4, \sqrt{dT/b}\}$.

### A.5   Proof of Thm. 5

The proof is rather involved, and is composed of several stages. First, we define a variant of our hide-and-seek problem, which depends on sparse distributions. We then prove an information-theoretic lower bound on the achievable performance for this hide-and-seek problem with information constraints. The bound is similar to Thm. 3, but without an explicit dependence on the bias[5] $\rho$. We then show how the lower bound can be strengthened in the specific case of $b$-memory online protocols. Finally, we use these ingredients in proving Thm. 5.

We begin by defining the following hide-and-seek problem, which differs from problem 2 in that the distribution is supported on sparse instances. It is again parameterized by a dimension $d$, bias $\rho$, and sample size $mn$:

**Definition 3** (Hide-and-seek Problem 2). *Consider the set of distributions $\{\Pr_j(\cdot)\}_{j=1}^d$ over $\{-\mathbf{e}_i, +\mathbf{e}_i\}_{i=1}^d$, defined as*

$$\Pr_j(\mathbf{e}_i) = \begin{cases} \frac{1}{2d} & i \neq j \\ \frac{1}{2d} + \frac{\rho}{d} & i = j \end{cases} \quad \Pr_j(-\mathbf{e}_i) = \begin{cases} \frac{1}{2d} & i \neq j \\ \frac{1}{2d} - \frac{\rho}{d} & i = j \end{cases}.$$

*Given an i.i.d. sample of $mn$ instances generated from $\Pr_j(\cdot)$, where $j$ is unknown, detect $j$.*

In words, $\Pr_j(\cdot)$ corresponds to picking $\pm\mathbf{e}_i$ where $i$ is chosen uniformly at random, and the sign is chosen uniformly if $i \neq j$, and positive (resp. negative) with probability $\frac{1}{2} + \rho$ (resp. $\frac{1}{2} - \rho$) if $i = j$. It is easily verified that this creates sparse instances with zero-mean coordinates, except coordinate $j$ whose expectation is $2\rho/d$.

We now present a result similar to Thm. 3 for this new hide-and-seek problem:

**Theorem 6.** *Consider hide-and-seek problem 2 on $d > 1$ coordinates, with some bias $\rho \leq \min\{\frac{1}{27}, \frac{1}{9\log(d)}, \frac{d}{14n}\}$. Then for any estimate $\tilde{J}$ of the biased coordinate returned by any $(b, n, m)$ protocol, there exists some coordinate $j$ such that*

$$\Pr_J(\tilde{J} = j) \leq \frac{3}{d} + 11\sqrt{\frac{mb}{d}}.$$

The proof appears in subsection A.6 below, and is broadly similar to the proof of Thm. 3 (although using a somewhat different approach).

The theorems above hold for any $(b, n, m)$ protocol, and in particular for $b$-memory online protocols (since they are a special case of $(b, 1, m)$ protocols). However, for $b$-online protocols, the following simple observation will allow us to further strengthen our results:

**Theorem 7.** *Any $b$-memory online protocol over $m$ instances is also a $\left(b, \kappa, \left\lfloor\frac{m}{\kappa}\right\rfloor\right)$ protocol for any positive integer $\kappa \leq m$.*

The proof is immediate: Given a a batch of $\kappa$ instances, we can always feed the instances one by one to our $b$-memory online protocol, and output the final message after $\lfloor m/\kappa \rfloor$ such batches are processed, ignoring any remaining instances. This makes the algorithm a type of $\left(b, \kappa, \lfloor \frac{m}{\kappa} \rfloor\right)$ protocol.

As a result, when discussing $b$-memory online protocols for some particular value of $m$, we can actually apply Thm. 6 where we replace $n, m$ by $\kappa, \lfloor m/\kappa \rfloor$, where $\kappa$ is a free parameter we can tune to attain the most convenient bound.

With these results at hand, we turn to prove Thm. 5.

The lower bound follows from the concentration of measure assumption on $\widetilde{x_i x_j}$, and a union bound, which implies that

$$\Pr\left(\forall i < j, \ |\widetilde{x_i x_j} - \mathbb{E}[x_i x_j]| < \frac{\tau}{2}\right) \geq 1 - \frac{d(d-1)}{2} 2\exp\left(-m\tau^2/6\right) \geq 1 - d^2 \exp\left(-m\tau^2/6\right).$$

If this event occurs, then picking $(\tilde{I}, \tilde{J})$ to be the coordinates with the largest empirical mean would indeed succeed in detecting $(i^*, j^*)$, since $\mathbb{E}[x_{i^*} x_{j^*}] \geq \mathbb{E}[x_i x_j] + \tau$ for all $(i, j) \neq (i^*, j^*)$.

The upper bound in the theorem statement follows from a reduction to the setting discussed in Thm. 6. Let $\{\Pr_{i^*, j^*}(\cdot)\}_{1 \leq i^* < j^* \leq d}$ be a set of distributions over $d$-dimensional vectors $\mathbf{x}$, parameterized by coordinate pairs $(i^*, j^*)$. Each such $\Pr_{i^*, j^*}(\cdot)$ is defined as a distribution over vectors of the form $\sqrt{\frac{d}{2}}(\sigma_1 \mathbf{e}_i + \sigma_2 \mathbf{e}_j)$ in the following way:

- $(i, j)$ is picked uniformly at random from $\{(i, j) : 1 \leq i < j \leq d\}$
- $\sigma_1$ is picked uniformly at random from $\{-1, +1\}$.
- If $(i, j) \neq (i^*, j^*)$, $\sigma_2$ is picked uniformly at random from $\{-1, +1\}$. If $(i, j) = (i^*, j^*)$, then $\sigma_2$ is chosen to equal $\sigma_1$ with probability $\frac{1}{2} + \rho$ (for some $\rho \in (0, 1/2)$ to be determined later), and $-\sigma_1$ otherwise.

In words, each instance is a 2-sparse random vector, where the two non-zero coordinates are chosen at random, and are slightly correlated if and only if those coordinates are $(i^*, j^*)$.

Let us first verify that any such distribution $\Pr_{i^*, j^*}(\cdot)$ belongs to the distribution family specified in the theorem:

1. For any coordinate $k$, $x_k$ is non-zero with probability $2/d$ (i.e. the probability that either $i$ or $j$ above equal $k$), in which case $x_k^2 = d/2$. Therefore, $\mathbb{E}[x_k^2] = 1$ for all $k$.
2. When $(i, j) \neq (i^*, j^*)$, then $\sigma_1, \sigma_2$ are uncorrelated, hence $\mathbb{E}[x_i x_j] = 0$. On the other hand, $\mathbb{E}[x_{i^*} x_{j^*}] = \frac{2}{d(d-1)}\left(\left(\frac{1}{2} + \rho\right)\frac{d}{2} + \left(\frac{1}{2} - \rho\right)\left(-\frac{d}{2}\right)\right) = \frac{2\rho}{d-1}$. So we can take $\tau = \frac{2\rho}{d-1}$, and have that $\mathbb{E}[x_{i^*} x_{j^*}] = \tau$.
3. For any $i < j$, $x_i x_j$ is a random variable which is non-zero with probability $2/(d(d-1))$, in which case its magnitude is $d/2$. Thus, $\mathbb{E}[(x_i x_j)^2] \leq \frac{d}{2(d-1)}$. Applying Bernstein's inequality, if $\widetilde{x_i x_j}$ is the empirical average of $x_i x_j$ over $m$ i.i.d. instances, then

$$\Pr\left(|\widetilde{x_i x_j} - \mathbb{E}[x_i x_j]| \geq \frac{\tau}{2}\right) \leq 2\exp\left(-\frac{m\tau^2}{4\left(\frac{d}{d-1} + \frac{d}{3}\tau\right)}\right).$$

Since we chose $\tau = \frac{2\rho}{d-1} < \frac{1}{d-1}$, and we assume $d \geq 9$, this bound is at most

$$2\exp\left(-\frac{m\tau^2}{\frac{4d}{d-1}\left(1 + \frac{1}{3}\right)}\right) \leq 2\exp\left(-\frac{m\tau^2}{6}\right).$$

Therefore, this distribution satisfies the theorem's conditions.

The crucial observation now is that the problem of detecting $(i^*, j^*)$ is can be reduced to a hide-and-seek problem as defined in Definition 3. To see why, let us consider the distribution over $d \times d$

matrices induced by $\mathbf{x}\mathbf{x}^\top$, where $\mathbf{x}$ is sampled according to $\mathrm{Pr}_{i^*,j^*}(\cdot)$ as described above, and in particular the distribution on the entries above the main diagonal. It is easily seen to be equivalent to a distribution which picks one entry $(i,j)$ uniformly at random from $\{(i,j) : 1 \le i < j \le d\}$, and assigns to it the value $\left\{-\frac{d}{2}, +\frac{d}{2}\right\}$ with equal probability, unless $(i,j) = (i^*,j^*)$, in which case the positive value is picked with probability $\frac{1}{2} + \rho$, and the negative value with probability $\frac{1}{2} - \rho$. This is equivalent to the hide-and-seek problem described in Definition 3, over $\frac{d(d-1)}{2}$ coordinates. Thus, we can apply Thm. 6 for $\frac{d(d-1)}{2}$ coordinates, and get that if $\rho \le \min\left\{\frac{1}{27}, \frac{1}{9\log\left(\frac{d(d-1)}{2}\right)}, \frac{d(d-1)}{28n}\right\}$, then for some $(i^*,j^*)$ and any estimator $(\tilde{I}, \tilde{J})$ returned by a $(b,n,m)$ protocol,

$$\mathrm{Pr}_{i^*,j^*}\left((\tilde{I}, \tilde{J}) = (i^*, j^*)\right) \le \frac{6}{d(d-1)} + 11\sqrt{\frac{2mb}{d(d-1)}}.$$

Our theorem deals with two types of protocols: $\left(b, d(d-1), \left\lfloor\frac{m}{d(d-1)}\right\rfloor\right)$ protocols, and $b$-memory online protocols over $m$ instances. In the former case, we can simply plug in $\left\lfloor\frac{m}{d(d-1)}\right\rfloor, d(d-1)$ instead of $m, n$, while in the latter case we can still replace $m, n$ by $\left\lfloor\frac{m}{d(d-1)}\right\rfloor, d(d-1)$ thanks to Thm. 7. In both cases, doing this replacement and choosing $\rho = \frac{1}{9\log\left(\frac{d(d-1)}{2}\right)}$ (which is justified when $d \ge 9$, as we assume), we get that

$$\mathrm{Pr}_{i^*,j^*}\left((\tilde{I}, \tilde{J}) = (i^*, j^*)\right) \le \frac{6}{d(d-1)} + 11\sqrt{\frac{2b}{d(d-1)}\left\lfloor\frac{m}{d(d-1)}\right\rfloor} \le \mathcal{O}\left(\frac{1}{d^2} + \sqrt{\frac{m}{d^4/b}}\right). \tag{19}$$

This implies the upper bound stated in the theorem, and also noting that

$$\tau = \frac{2\rho}{d-1} = \frac{2}{9(d-1)\log\left(\frac{d(d-1)}{2}\right)} = \Theta\left(\frac{1}{d\log(d)}\right).$$

Having finished with the proof of the theorem as stated, we note that it is possible to extend the construction used here to show performance gaps for other sample sizes $m$. For example, instead of using a distribution supported on

$$\left\{\sqrt{\frac{d}{2}}\left(\sigma_1\mathbf{e}_i + \sigma_2\mathbf{e}_j\right)\right\}_{1 \le i < j \le d}$$

for any pair of coordinates $1 \le i < j \le d$, we can use a distribution supported on

$$\left\{\sqrt{\frac{\lambda}{2}}\left(\sigma_1\mathbf{e}_i + \sigma_2\mathbf{e}_j\right)\right\}_{1 \le i < j \le \lambda}$$

for some $\lambda \le d$. By choosing the bias $\tau = \Theta(1/\lambda\log(\lambda))$, we can show a performance gap (in detecting the correlated coordinates) in the regime $\frac{\lambda^4}{b} \gg m \gg \lambda^2\log^2(\lambda)$. This regime exists for $\lambda$ as small as $\sqrt{b}$ (up to log-factors), in which case we already get performance gaps when $m$ is roughly linear in the memory $b$.

## A.6 Proof of Thm. 6

The proof builds on the auxiliary lemmas presented in Appendix A.1. It is broadly similar to the proof of Thm. 3, but with a few more technical intricacies (such as balls-and-bins arguments) to handle the different sampling process.

On top of the distributions $\mathrm{Pr}_j(\cdot)$ defined in the hide-and-seek problem (Definition 3), we define an additional 'reference' distribution $\mathrm{Pr}_0(\cdot)$, which corresponds to the instances being chosen uniformly at random from $\{-\mathbf{e}_i, +\mathbf{e}_i\}_{i=1}^d$ (i.e. there is no biased coordinate).

Let $w^1, \ldots, w^m$ denote the messages computed by the protocol. To show the lower bound, it is enough to prove that

$$\frac{2}{d} \sum_{j=1}^{d} D_{kl} \left( \text{Pr}_0(w^1 \ldots w^m) || \text{Pr}_j(w^1 \ldots w^m) \right) \leq \frac{26mb}{d}, \qquad (20)$$

since then by applying Lemma 1, we get that for some $j$, $\text{Pr}_j(\tilde{J} = j) \leq (3/d) + 2\sqrt{26mb/d} < (3/d) + 11\sqrt{mb/d}$ as required.

Using the chain rule, the left hand side in Eq. (20) equals

$$\frac{2}{d} \sum_{j=1}^{d} \sum_{t=1}^{m} \mathbb{E}_{w^1 \ldots w^{t-1} \sim \text{Pr}_0} \left[ D_{kl} \left( \text{Pr}_0(w^t | w^1 \ldots w^{t-1}) || \text{Pr}_j(w^t | w^1 \ldots w^{t-1}) \right) \right]$$

$$= 2 \sum_{t=1}^{m} \mathbb{E}_{w^1 \ldots w^{t-1} \sim \text{Pr}_0} \left[ \frac{1}{d} \sum_{j=1}^{d} D_{kl} \left( \text{Pr}_0(w^t | w^1 \ldots w^{t-1}) || \text{Pr}_j(w^t | w^1 \ldots w^{t-1}) \right) \right] \qquad (21)$$

Let us focus on a particular choice of $t$ and values $w^1 \ldots w^{t-1}$. To simplify the presentation, we drop the $^t$ superscript from the message $w^t$, and denote the previous messages $w^1 \ldots w^{t-1}$ as $\hat{w}$. Thus, we consider the quantity

$$\frac{1}{d} \sum_{j=1}^{d} D_{kl} \left( \text{Pr}_0(w|\hat{w}) || \text{Pr}_j(w|\hat{w}) \right). \qquad (22)$$

Recall that $w$ is some function of $\hat{w}$ and a set of $n$ instances received in the current round. Moreover, each instance is non-zero at a single coordinate, with a value in $\{-1, +1\}$. Thus, given an ordered sequence of $n$ instances, we can uniquely specify them using vectors $\mathbf{u}, \mathbf{x}_1, \ldots, \mathbf{x}_d$, where

- $\mathbf{u} \in \{1 \ldots d\}^n$, where each $e_i$ indicates what is the non-zero coordinate of the $i$-th instance.
- Each $\mathbf{x}_j \in \{-1, +1\}^{|\{i:e_i=j\}|}$ is a (possibly empty) vector of the non-zero values, when those values fell in coordinate $j$.

For example, if $d = 3$ and the instances are $(-1, 0, 0); (0, 1, 0); (0, -1, 0)$, then $\mathbf{u} = (1, 2, 2); \mathbf{x}_1 = (-1); \mathbf{x}_2 = (1, -1); \mathbf{x}_3 = \emptyset$. Note that under both $\text{Pr}_0(\cdot)$ and $\text{Pr}_j(\cdot)$, $\mathbf{u}$ is uniformly distributed in $\{1 \ldots d\}^n$, and $\{\mathbf{x}_j\}_j$ are mutually independent conditioned on $\mathbf{u}$.

With this notation, we can rewrite Eq. (22) as

$$\frac{1}{d} \sum_{j=1}^{d} D_{kl} \left( \mathbb{E}_{\mathbf{u}} \left[ \text{Pr}_0(w|\mathbf{u}, \hat{w}) \right] || \mathbb{E}_{\mathbf{u}} \left[ \text{Pr}_j(w|\mathbf{u}, \hat{w}) \right] \right).$$

Since the relative entropy is jointly convex in its arguments, we have by Jensen's inequality that this is at most

$$\frac{1}{d} \sum_{j=1}^{d} \mathbb{E}_{\mathbf{u}} \left[ D_{kl} \left( \text{Pr}_0(w|\mathbf{u}, \hat{w}) || \text{Pr}_j(w|\mathbf{u}, \hat{w}) \right) \right] = \mathbb{E}_{\mathbf{u}} \left[ \frac{1}{d} \sum_{j=1}^{d} D_{kl} \left( \text{Pr}_0(w|\mathbf{u}, \hat{w}) || \text{Pr}_j(w|\mathbf{u}, \hat{w}) \right) \right].$$
$$(23)$$

We now decompose $\text{Pr}_j(w|\mathbf{u}, \hat{w})$ using Lemma 2, where $p(\cdot) = \text{Pr}_j(\cdot|\mathbf{u}, \hat{w}), q(\cdot) = \text{Pr}_0(\cdot|\mathbf{u}, \hat{w})$ and each $z_i$ is $\mathbf{x}_i$. The lemma's conditions are satisfied since the distribution of $\mathbf{x}_i$, $i \neq j$ is the same under $\text{Pr}_0(\cdot|\mathbf{u}, \hat{w}), \text{Pr}_j(\cdot|\mathbf{u}, \hat{w})$, and also $w$ is only a function of $\mathbf{u}, \mathbf{x}_1 \ldots \mathbf{x}_d$ and $\hat{\mathbf{w}}$. Thus, we can rewrite Eq. (23) as

$$\mathbb{E}_{\mathbf{u}} \left[ \frac{1}{d} \sum_{j=1}^{d} D_{kl} \left( \text{Pr}_0(w|\mathbf{u}, \hat{w}) \,\middle|\middle|\, \sum_{\mathbf{x}_j} \text{Pr}_j(\mathbf{x}_j|\mathbf{u}, \hat{w}) \text{Pr}_0(w|\mathbf{u}, \mathbf{x}_j, \hat{w}) \right) \right].$$

Using Lemma 3, we can reverse the expressions in the relative entropy term, and upper bound the above by

$$\mathbb{E}_{\mathbf{u}}\left[\frac{1}{d}\sum_{j=1}^{d}\left(\max_{w}\frac{\Pr_0(w|\mathbf{u},\hat{w})}{\sum_{\mathbf{x}_j}\Pr_j(\mathbf{x}_j|\mathbf{u},\hat{w})\Pr_0(w|\mathbf{u},\mathbf{x}_j,\hat{w})}\right)\times\right.$$
$$\left. D_{kl}\left(\sum_{\mathbf{x}_j}\Pr_j(\mathbf{x}_j|\mathbf{u},\hat{w})\Pr_0(w|\mathbf{u},\mathbf{x}_j,\hat{w})\;\middle\|\;\Pr_0(w|\mathbf{u},\hat{w})\right)\right]. \tag{24}$$

The max term equals

$$\max_{w}\frac{\sum_{\mathbf{x}_j}\Pr_0(\mathbf{x}_j|\mathbf{u},\hat{w})\Pr_0(w|\mathbf{u},\mathbf{x}_j,\hat{w})}{\sum_{\mathbf{x}_j}\Pr_j(\mathbf{x}_j|\mathbf{u},\hat{w})\Pr_0(w|\mathbf{u},\mathbf{x}_j,\hat{w})}\leq\max_{\mathbf{x}_j}\frac{\Pr_0(\mathbf{x}_j|\mathbf{u},\hat{w})}{\Pr_j(\mathbf{x}_j|\mathbf{u},\hat{w})},$$

and using Jensen's inequality and the fact that relative entropy is convex in its arguments, we can upper bound the relative entropy term by

$$\sum_{\mathbf{x}_j}\Pr_j(\mathbf{x}_j|\mathbf{u},\hat{w})D_{kl}\left(\Pr_0(w|\mathbf{u},\mathbf{x}_j,\hat{w})\;\|\;\Pr_0(w|\mathbf{u},\hat{w})\right)$$
$$\leq\left(\max_{\mathbf{x}_j}\frac{\Pr_j(\mathbf{x}_j|\mathbf{u},\hat{w})}{\Pr_0(\mathbf{x}_j|\mathbf{u},\hat{w})}\right)\sum_{\mathbf{x}_j}\Pr_0(\mathbf{x}_j|\mathbf{u},\hat{w})D_{kl}\left(\Pr_0(w|\mathbf{u},\mathbf{x}_j,\hat{w})\;\|\;\Pr_0(w|\mathbf{u},\hat{w})\right).$$

The sum in the expression above equals the mutual information between the message $w$ and the coordinate vector $\mathbf{x}_j$ (seen as random variables with respect to the distribution $\Pr_0(\cdot|\mathbf{u},\hat{w})$). Writing this as $I_{\Pr_0(\cdot|\mathbf{u},\hat{w})}(w;\mathbf{x}_j)$, we can thus upper bound Eq. (24) by

$$\mathbb{E}_{\mathbf{u}}\left[\frac{1}{d}\sum_{j=1}^{d}\left(\max_{\mathbf{x}_j}\frac{\Pr_0(\mathbf{x}_j|\mathbf{u},\hat{w})}{\Pr_j(\mathbf{x}_j|\mathbf{u},\hat{w})}\right)\left(\max_{\mathbf{x}_j}\frac{\Pr_j(\mathbf{x}_j|\mathbf{u},\hat{w})}{\Pr_0(\mathbf{x}_j|\mathbf{u},\hat{w})}\right)I_{\Pr_0(\cdot|\mathbf{u},\hat{w})}(w;\mathbf{x}_j)\right]$$
$$\leq\mathbb{E}_{\mathbf{u}}\left[\left(\max_{j,\mathbf{x}_j}\frac{\Pr_0(\mathbf{x}_j|\mathbf{u},\hat{w})}{\Pr_j(\mathbf{x}_j|\mathbf{u},\hat{w})}\right)\left(\max_{j,\mathbf{x}_j}\frac{\Pr_j(\mathbf{x}_j|\mathbf{u},\hat{w})}{\Pr_0(\mathbf{x}_j|\mathbf{u},\hat{w})}\right)\frac{1}{d}\sum_{j=1}^{d}I_{\Pr_0(\cdot|\mathbf{u},\hat{w})}(w;\mathbf{x}_j)\right].$$

Since $\mathbf{x}_1\ldots\mathbf{x}_d$ are mutually independent conditioned on $\mathbf{u}$ and $\hat{w}$, and also $w$ contains at most $b$ bits, we can use the key Lemma 6 to upper bound the above by

$$\mathbb{E}_{\mathbf{u}}\left[\left(\max_{j,\mathbf{x}_j}\frac{\Pr_0(\mathbf{x}_j|\mathbf{u},\hat{w})}{\Pr_j(\mathbf{x}_j|\mathbf{u},\hat{w})}\right)\left(\max_{j,\mathbf{x}_j}\frac{\Pr_j(\mathbf{x}_j|\mathbf{u},\hat{w})}{\Pr_0(\mathbf{x}_j|\mathbf{u},\hat{w})}\right)\frac{b}{d}\right].$$

Now, recall that conditioned on $\mathbf{u}$, each $\mathbf{x}_j$ refers to a column of $|\{i:e_i=j\}|$ i.i.d. entries, drawn independently of any previous messages $\hat{w}$, where under $\Pr_0$, each entry is chosen to be $\pm 1$ with equal probability, whereas under $\Pr_j$ each is chosen to be $1$ with probability $\frac{1}{2}+\rho$, and $-1$ with probability $\frac{1}{2}-\rho$. Therefore, we can upper bound the expression above by

$$\mathbb{E}_{\mathbf{u}}\left[\left(\max_{j}\max\left\{\left(\frac{1/2+\rho}{1/2}\right)^{|\{i:e_i=j\}|},\left(\frac{1/2}{1/2-\rho}\right)^{|\{i:e_i=j\}|}\right\}\right)^2\frac{b}{d}\right].$$

Since we assume $\rho\leq 1/27$, it's easy to verify that the expression above is at most

$$\mathbb{E}_{\mathbf{u}}\left[\left(\max_{j}(1+2.2\rho)^{|\{i:e_i=j\}|}\right)^2\frac{b}{d}\right]=\mathbb{E}_{\mathbf{u}}\left[\left(\max_{j}\left(1+\frac{4.4\rho|\{i:e_i=j\}|}{2|\{i:e_i=j\}|}\right)^{2|\{i:e_i=j\}|}\right)\frac{b}{d}\right]$$
$$\leq\mathbb{E}_{\mathbf{u}}\left[\max_{j}\exp\left(4.4\rho|\{i:e_i=j\}|\right)\right]\frac{b}{d}=\mathbb{E}_{\mathbf{u}}\left[\exp\left(4.4\rho\max_{j}|\{i:e_i=j\}|\right)\right]\frac{b}{d}$$

Since $\mathbf{u}$ is uniformly distributed in $\{1\ldots d\}^n$, then $\max_j|\{i:e_i=j\}$ corresponds to the largest number of balls in a bin when we randomly throw $n$ balls into $d$ bins. By Lemma 5, and since we assume $\rho\leq\min\{\frac{1}{27},\frac{1}{9\log(d)},\frac{d}{14n}\}$, it holds that the expression above is at most $13b/d$. To summarize, this is a valid upper bound on Eq. (22), i.e. on any individual term inside the expectation in Eq. (21). Thus, we can upper bound Eq. (21) by $26mb/d$. This shows that Eq. (20) indeed holds, which as explained earlier implies the required result.

# B  Basic Results in Information Theory

The proof of Thm. 3 and Thm. 6 makes extensive use of quantities and basic results from information theory. We briefly review here the technical results relevant for our paper. A more complete introduction may be found in [14]. Following the settings considered in the paper, we will focus only on discrete distributions taking values on a finite set.

Given a random variable $X$ taking values in a domain $\mathcal{X}$, and having a distribution function $p(\cdot)$, we define its entropy as

$$H(X) = \sum_{x \in \mathcal{X}} p(x) \log_2(1/p(x)) = \mathbb{E}_X \log_2\left(\frac{1}{p(x)}\right).$$

Intuitively, this quantity measures the uncertainty in the value of $X$. This definition can be extended to joint entropy of two (or more) random variables, e.g. $H(X,Y) = \sum_{x,y} p(x,y) \log_2(1/p(x,y))$, and to conditional entropy

$$H(X|Y) = \sum_y p(y) \sum_x p(x|y) \log_2\left(\frac{1}{p(x|y)}\right).$$

For a particular value $y$ of $Y$, we have

$$H(X|Y=y) = \sum_x p(x|y) \log_2\left(\frac{1}{p(x|y)}\right)$$

It is possible to show that $\sum_{j=1}^n H(X_i) \geq H(X_1, \ldots, X_n)$, with equality when $X_1, \ldots, X_n$ are independent. Also, $H(X) \geq H(X|Y)$ (i.e. conditioning can only reduce entropy). Finally, if $X$ is supported on a discrete set of size $2^b$, then $H(X)$ is at most $b$.

Mutual information $I(X;Y)$ between two random variables $X, Y$ is defined as

$$I(X;Y) = H(X) - H(X|Y) = H(Y) - H(Y|X) = \sum_{x,y} p(x,y) \log_2\left(\frac{p(x,y)}{p(x)p(y)}\right).$$

Intuitively, this measures the amount of information each variable carries on the other one, or in other words, the reduction in uncertainty on one variable given we know the other. Since entropy is always positive, we immediately get $I(X;Y) \leq \min\{H(X), H(Y)\}$. As for entropy, one can define the conditional mutual information between random variables X,Y given some other random variable $Z$ as

$$I(X;Y|Z) = \mathbb{E}_{z \sim Z}[I(X;Y|Z=z)] = \sum_z p(z) \sum_{x,y} p(x,y|z) \log_2\left(\frac{p(x,y|z)}{p(x|z)p(y|z)}\right).$$

Finally, we define the relative entropy (or Kullback-Leibler divergence) between two distributions $p, q$ on the same set as

$$D_{kl}(p||q) = \sum_x p(x) \log_2\left(\frac{p(x)}{q(x)}\right).$$

It is possible to show that relative entropy is always non-negative, and jointly convex in its two arguments (viewed as vectors in the simplex). It also satisfies the following chain rule:

$$D_{kl}(p(x_1 \ldots x_n)||q(y_1 \ldots y_n)) = \sum_{i=1}^n \mathbb{E}_{x_1 \ldots x_{i-1} \sim p}\left[D_{kl}(p(x_i|x_1 \ldots x_{i-1})||q(x_i|x_1 \ldots x_{i-1}))\right].$$

Also, it is easily verified that

$$I(X;Y) = \sum_y p(y) \, D_{kl}(p_X(\cdot|y)||p_X(\cdot)),$$

where $p_X$ is the distribution of the random variable $X$. In addition, we will make use of Pinsker's inequality, which upper bounds the so-called total variation distance of two distributions $p, q$ in terms of the relative entropy between them:

$$\sum_x |p(x) - q(x)| \leq \sqrt{2D_{kl}(p||q)}.$$

Finally, an important inequality we use in the context of relative entropy calculations is the log-sum inequality. This inequality states that for any nonnegative $a_i, b_i$,

$$\left(\sum_i a_i\right) \log \frac{\sum_i a_i}{\sum_i b_i} \leq \sum_i a_i \log \frac{a_i}{b_i}.$$

## Footnotes

[3]We suspect that this construction can be simplified, but were unable to achieve this without considerably weakening the bound.

[4]The theorem discusses the case where the distribution is over $\{-1, +1\}^d$, and coordinate $j$ has a slight positive bias, but it's easily seen that the lower bound also holds here where the domain is $\{0, 1\}^d$.

[5]Attaining a dependence on $\rho$ seems technically complex for this hide-and-seek problem, but fortunately is not needed to prove Thm. 5.