[Reviews · NeurIPS 2014]

Submitted by Assigned_Reviewer_25

The paper considers the effect of memory constraints on some classes of online and distributed algorithms. The main goal of the authors is to show that there exist learning problems where imposing a memory constraint provably hurts the performance of any algorithm, in the sense that the best achievable performance guarantees are worse than some known upper bound on an algorithm that operates without memory constraints. The authors also provide a lower bound on the regret of online algorithms that operate under a specific feedback constraint. The results of the paper follow from an elegant information-theoretic argument concerning "hide-and-seek problems" where a learner has to detect a biased coordinate of i.i.d. random vectors by sequentially sampling a bounded number of coordinates.

The paper is written very well and the technical quality is excellent. To my best knowledge, the main results of the paper (Theorems 2 and 3) are novel, and so are their implications in the considered learning settings. My main problem is that these settings are not general enough to draw significant conclusions as done by the authors. Generally, I find that the presented results are a bit oversold, and the paper would benefit from a more modest presentation. I list my main observations considering the contributions below.

First, the lower bounds on the regret achievable by memory-bounded online learning algorithms seem to directly follow from the results of Seldin, Bartlett, Crammer and Abbasi-Yadkori (ICML 2014), whose counterexample is identical to the one presented in this paper. While this alone would not be a problem, it seems that the bound presented in Theorem 4 is only meaningful when the adversary is restricted to play binary losses (or at least choose from a finite pool of loss vectors). Indeed, allowing the learner to observe \Omega(d\sqrt{T}) bits of the loss of a *fixed* arm in each round is clearly not enough for any algorithm to achieve consistency, yet the lower bound of Theorem 4 fails to capture this intuition as it only guarantees a regret of \Omega(1). Theorem 4 is also clearly loose in finite-outcome partial monitoring games with minimax regret of \Theta(T^{2/3}). That is, the claim that Theorem 4 "quantifies in a very direct way the cost of information constraints for such learning problems" (line 303) is quite misleading.

The examples considered in Sections 4.2 and 4.3 theoretically support the intuition that some learning problems are difficult to solve if the learner is forced to work with a memory budget that is significantly smaller than the natural representation of the problem instance to learn.
Although it is nice to see such a result proven formally, this is hardly surprising: The same kind of information vs. performance trade-off can be observed when comparing the minimax performance guarantees concerning multi-armed-bandit and online prediction problems. While this contribution is somewhat stronger than the results of Section 4.1, I don't feel that the lesson is as fundamental as the authors claim.

Detailed comments
-----------------

159: "Let X^t be a batch of i.i.d. instances" -- instances of precisely what?
172,173: correspond -> corresponds
311: It would be interesting to contrast these results to those of Van Erven, Kotlowski and Warmuth (COLT, 2014), who propose a learning algorithm based on dropping out elements of each loss vector with an arbitrary fixed probability. When setting the dropout probability sufficiently high, one can guarantee that at most b bits are retained in each round with high probability, yet their algorithm succeeds in obtaining regret guarantees of O(sqrt{T log d}). I wonder if this contradicts the results of Theorem 4...?
384: 1 - exp(-\Omega(...)) -> \Omega(1-exp(-...))
407: While any b-memory protocol is indeed a (b,k,[m/k]) protocol, the converse is not true, thus this reduction doesn't seem to work.
678, 825: x_j is missing from the conditions of Pr_0(.|.)
Summary: While the paper raises some valid issues, and provides partial answers by applying an elegant technique, the results are not as striking as the authors claim. The paper could be made stronger by considering more interesting instantiations of the main results of Thms 2 and 3.

Submitted by Assigned_Reviewer_30

SUMMARY

This paper provides new information-theoretic limits on performance of a wide variety of learning algorithms in stochastic i.i.d. settings. It presents a simple problem ('hide and seek') and a deceptively simple-looking result (Theorem 2 and 3), providing an upper bound on the probability of correctly guessing a distribution
when m batches of n data points are observed and only b bits can be extracted from any batch, yet the function mapping the n data points to a b-bit representation can be chosen any way one likes, even adaptively.

QUALITY

The result is useful: Theorem 2 and 3 are easy to understand and its application to a variety of learning problems is easy, even trivial. But the proof of theorems 2-3 seems highly nontrivial.
At first I thought the results would follow straightforwardly using e.g. Fano's theorem (an information theorist's first resort for problems like this) or other techniques taken from the minimax-lower-bound-literature on adaptive nonparametric estimation (such as Yang and Barron's paper on minimax rates). But upon closer inspection this does not seem to be the case, one really needs a much more involved proof, making the result more interesting.

My one issue with the paper - but this is a very serious one - is that the proof of theorems 2-3 takes 9 pages, all of which is in the appendix which I am not even required to read! This is especially disturbing since the results by themselves do not look very surprising; for me, they mostly become interesting because they do not follow straightforwardly from existing well-known results, so the interesting (for me) part of the paper is in how to prove these results. Then one really wants to see more of this in the main text.

Now, the main text does give a nice overview of what the difficulties in the proof are, but does not give any high-level overview of the ideas used to tackle these difficulties (the one thing that is mentioned, recent relations between KL divergence, its reverse form and Xi2 divergence, is more or less straightforward, to me that's really not the real difficulty). Thus, this paper really relies for 100% on its appendix/additional material which is longer than the paper itself. I am torn between simply saying 'clear reject - because neither me nor, I suspect, any of the other reviewers will have looked at the proofs in any detail, so we cannot have this at NIPS - submit this to a journal instead' or 'clear accept', an 8, - let's trust the authors because superficially it all looks solid and the results are useful'. in the end I voted a 6 on quality as the average between 4 and 8. I guess the area/PC chairs have to make a real decision on this.

In the mean time, a question that came up: can't Fano's inequality be used after all? I tried for a while and saw that it isn't trivial at all, but it may still avoid some of the complications of what you are doing; intuitively, it should be sufficient to get a good bound on the conditional entropy H(J |\tilde{J}), where we assume J, the nonuniform coordinate, is selected uniformly at random from the d choices.

CLARITY

Generally quite clear. There's something slightly wrong with the notation in the second paragraph in Section 4.3 (v_1, v_2 -> shouldn't that be v_i, v_j ? and x_i -> why the boldface?)

ORIGINALITY

As far as I can judge, highly original.

SIGNIFICANCE

Opens up new areas of research, so yes, pretty significant.

UPDATE
After reading the author's response I updated my score 7, since they indeed have a point that there are many similar papers with all proofs in the supplementary material. Having said that I'm still not 100% happy with it in this particular case because with this paper, for me 'the proof of the pudding is in the proof' so to speak, not in the eating - the proof is perhaps the most interesting part.
Summary: Original and useful paper on information-theoretic bounds in learning under memory or communication constraints; but it is problematic that the highly nontrivial and very long proof finds itself 100% in the additional material.

Submitted by Assigned_Reviewer_37

The paper is about understanding capabilities of various machine learning
algorithms subject to various kinds of constraints: communication (in
distributed settings), memory, online vs batch, and partial access.

The main results are based on a problem that the authors call as
"hide-and-seek" problem, but is basically finding one object out of many
(n), where the particular object is a bit that has a bias, whereas all
others are unbiased. This problem is defined with respect to product
distributions over n-bit boolean vectors. The authors then argue that
resource-bounded algorithms (of various kinds) cannot solve this problem
with the same efficiency as resource unbounded algorithms (all these are
information-theoretic results).
Summary: The paper is about understanding capabilities of various machine learning
algorithms subject to various kinds of constraints: communication (in
distributed settings), memory, online vs batch, and partial access. It
shows information-theoretic limitations on algorithms operating with
limited resources when compared to those operating with unbounded
resources.
Author Feedback
Author rebuttal: We thank all reviewers for their comments.

Review 1 (assigned reviewer 25)
===============================
General note: We're sorry to read that the reviewer thought our presentation insufficiently modest - this was definitely not our intention! We think the responses below should address his/her concerns, but in any case, would be glad to re-consider any statements in the text which the reviewer thinks are misleading.

Comment: Our regret lower bounds for memory-bounded algorithms follow directly from the results of Seldin et al. 2014.

Response:
After re-checking, we are quite certain that our bounds don't follow from that paper. Seldin et al. provide interesting and useful lower bounds, but they focus on a specific information feedback model, where the player pays to observe a subset of the losses (this is crucially used, for example, in the proof of theorem 2 in their paper, third step, where the KL divergence can be conveniently written as a fixed KL term times an expectation of an indicator to whether a given loss was observed). In contrast, our results (e.g. theorem 4) hold for any setting where the player observes a bounded number of bits: It can be a subset of the losses, but also a linear or non-linear projection of them, or arbitrary feedback signals of bounded size. To bound the relevant information divergences in this general setting is not trivial, and requires developing information contraction tools as described in our paper. See also review 2 which notes how our results do not follow from standard tools such as Fano's theorem or other minimax lower bound techniques.

Comment: Our lower bound in theorem 4 is only shown for discrete losses, and unlike what is implied by the text, it's not tight for some partial-information settings where the regret is no better than T^{2/3}.

Response: We thank the reviewer for the comment and we will amend the text to make it more precise. It is indeed true that we don't capture the optimal regret for any individual partial-information setting. However, as mentioned earlier, our focus is lower bounds which hold for *any* partial information model falling into our framework. Since in some models we *can* get T^{1/2} regret, we cannot hope to get T^{2/3} lower bounds without further assumptions. Regarding discrete losses, note that in practice any representation of the losses is finite-precision and hence discrete. We could extend our framework to continuous- valued losses, but then things get messy since standard bits are no longer the right tool to measure information here (e.g. even one continuous-valued loss can require an infinite number of bits). We will add a discussion of this issue.

Comment: The bounds shown for sparse-PCA and stochastic optimization are not surprising, since a similar kind of information vs. performance trade-off can be observed in multi-armed bandits.

Response: Of course, this is a valid subjective judgement, but let us explain again why we consider the information trade-offs here as qualitatively different than multi-armed bandits:
- First, they pertain to memory or distributed access constraints rather than partial-information constraints of a specific form.
- Second, in the sparse PCA case, the results are applicable even when the available memory is much *larger* than the natural representation of each training instance (as much as quadratic). In contrast, multi-armed bandits are in a setting where we can only remember/view a *small* part of each instance.
- Third, we are quite certain that our results cannot be obtained via a straightforward reduction to multi-armed bandits (see first comment above), and we're really not familiar with previous such trade-offs shown for these settings. If the reviewer is familiar with any (not discussed in the paper), we'd be happy to stand corrected.

Review 2 (assigned reviewer 30)
===============================

The reviewer's main concern appears to be that all the proofs are in the supplementary (otherwise, he would give a score of 8).
We thank the reviewer for this important comment. We added a high-level description of the proof, but agree it doesn't provide much technical detail. We will gladly add a more technical proof sketch in the final version, which will satisfy those interested in our techniques. But more generally, we'd like to point out that NIPS papers often leave all the proofs in the appendix, and sometimes don't even provide proof sketches. This even includes papers selected for oral presentations: Just for example, "From Bandits to Experts: A Tale of Domination and Independence" by Alon et al. from NIPS 2013, and "A Stochastic Gradient Method with an Exponential Convergence Rate for Finite Training Sets" by Le Roux et. al. from NIPS 2012.
One can argue whether this situation is desirable, but we think it would be unfair to judge our paper by different standards than other papers - especially when the reviewers do find our results original and significant. In any case, since we can add a more technical proof sketch, we think this should not be a major concern.